# BRANCH-AND-BOUND SEARCH FOR EXACT MAP INFERENCE IN CREDAL NETWORKS

**Radu Marinescu**
IBM Research
Dublin, Ireland
`radu.marinescu@ie.ibm.com`

**Fabio Cozman & Denis Maua**
University of Sao Paolo
Sao Paolo, Brazil
`{fgcozman,denis.maua}@usp.br`

**Debarun Bhattacharjya & Junkyu Lee**
IBM Research
Yorktown Heights, NY 10598, USA
`debarunb@us.ibm.com, junkyu.lee@ibm.com`

**Alexander Gray**
Centaur AI Institute
`alexander.gray@centaurinstitute.org`

## ABSTRACT

Credal networks extend Bayesian networks by incorporating imprecise probabilities through convex sets of probability distributions known as credal sets. MAP inference in credal networks, which seeks the most probable variable assignment given evidence, becomes inherently more difficult than in Bayesian networks because it involves computations over a complex joint credal set. In this paper, we introduce two tasks called *maximax* and *maximin* MAP, and develop depth-first branch-and-bound search algorithms for solving them *exactly*. The algorithms exploit problem decomposition by exploring an AND/OR search space and use a partitioning-based heuristic function enhanced with a cost-shifting scheme to effectively guide the search. Our experimental results obtained on both random and realistic credal networks clearly demonstrate the effectiveness of the proposed algorithms as they scale to large and complex problem instances.

## 1 INTRODUCTION

Credal networks (Cozman, 2000) are probabilistic graphical models that generalize Bayesian networks (Pearl, 1988) by allowing imprecise probabilities. Instead precise probability mass functions, they utilize convex sets of probability distributions known as credal sets to represent the local models for network variables given their parents. This enables a more flexible and robust treatment of uncertainty compared with Bayesian networks, accommodating severe uncertainty, unreliable data or conflicting information (Mauá & Cozman, 2020). Credal networks are especially valuable when precise probability estimates are difficult or undesirable to obtain. Moreover, credal networks are obtained in partially identifiable structural causal models with non-observed latent variables, as often met in causal discovery and inference (Zaffalon et al., 2020).

Over the past decades, research has primarily focused on developing marginal inference algorithms to efficiently compute the upper and lower probability bounds of a query variable given evidence in a credal network (Mauá & Cozman, 2020; Cano et al., 2007; Antonucci et al., 2010; Wijk et al., 2022). Maximum a Posteriori or MAP inference tasks for credal networks, which aim to identify

the most probable value assignments to the variables given evidence, have received relatively little attention from the community. This stands in stark contrast to Bayesian network MAP, which has been extensively investigated over the years (Koller & Friedman, 2009).

MAP inference in credal networks is substantially more challenging than in Bayesian networks due to computations over the joint credal set. Despite its difficulty, it remains relevant for explaining evidence, whether or not hidden variables are involved. Some recent work has proposed a variety of exact and approximate algorithms for Marginal MAP inference in credal networks, including variable elimination, exhaustive depth-first search and stochastic local search (Marinescu et al., 2023). Although these methods can be trivially extended to credal MAP inference they often scale poorly, limiting applicability to small models or offering no guarantees on solution quality.

**Contributions:**  This paper advances recent research on MAP inference in credal networks. In particular, we focus on two MAP tasks called *maximax* and *maximin* MAP, defined as finding an assignment to the network variables that is consistent with the evidence and has a maximum *upper* and, respectively, *lower probability*. We introduce new depth-first branch-and-bound search algorithms for solving these tasks *exactly* in practice. The methods leverage an AND/OR search space associated with the credal network, effectively exploiting the underlying problem structure during search. The proposed AND/OR search for credal networks extends the approach previously developed for Bayesian networks (Dechter & Mateescu, 2007). Furthermore, we enhance these algorithms with a novel partitioning-based heuristic that combines potential approximations with cost-shifting strategies to produce effective search heuristics. We empirically evaluate the new MAP inference algorithms on random credal networks with various graph topologies and on a collection of credal networks derived from real-world applications. The experimental results demonstrate that our algorithms significantly improve computational efficiency, scaling effectively to large problems with over 3000 variables while guaranteeing the optimality of the solutions found. Thus, our proposed approach addresses two major shortcomings of previous methods for MAP inference in credal networks: the lack of solution quality guarantees and the inability to solve large and complex problems. The Appendix includes additional details, experimental results, code and benchmarks.

## 2 BACKGROUND

### 2.1 BAYESIAN NETWORKS

A *Bayesian network* (BN) (Pearl, 1988) is defined by a tuple $\langle \mathbf{X}, \mathbf{D}, \mathbf{P}, G \rangle$, where $\mathbf{X} = \{X_1, \ldots, X_n\}$ is a set of variables over multi-valued domains $\mathbf{D} = \{D_1, \ldots, D_n\}$, $G$ is a directed acyclic graph (DAG) over $\mathbf{X}$ as nodes, and $\mathbf{P} = \{P_i\}$ where $P_i = P(X_i|\Pi_i))$ are *conditional probability tables* (CPTs) associated with each variable $X_i$ and $\Pi_i \subseteq \mathbf{X}$ are the parents of $X_i$ in $G$. A Bayesian network represents a joint probability distribution over $\mathbf{X}$, given by $P(\mathbf{X}) = \prod_{i=1}^{n} P(X_i|\Pi_i)$.

Given evidence $\mathbf{e}$ on a subset of variables $\mathbf{E} \subseteq \mathbf{X}$, the MAP task seeks an assignment $\mathbf{y}^* = (y_1^*, \ldots, y_m^*)$ to the remaining variables $\mathbf{Y} = \mathbf{X} \setminus \mathbf{E}$ that has a maximum probability:

$$\mathbf{y}^* = \arg\max_{\mathbf{y} \in \Omega(\mathbf{Y})} P(\mathbf{y}, \mathbf{e}) = \arg\max_{\mathbf{y} \in \Omega(\mathbf{Y})} \prod_{i=1}^{n} P(x_i|\pi_i) \tag{1}$$

where $\Omega(\mathbf{Y})$ denotes the Cartesian product of the domains of the variables in $\mathbf{Y}$, while $x_i$ and $\pi_i$ are the configurations of $X_i$ and $X_i$'s parents $\Pi_i$ in the assignment $\mathbf{x} = (\mathbf{y}, \mathbf{e})$ consistent with $\mathbf{e}$.

MAP is known to be NP-hard in general (Shimony, 1994; Kwisthout, 2011). However, in recent decades, several algorithmic schemes have been developed to solve MAP exactly (Kask & Dechter, 1999; Larrosa & Schiex, 2003; Marinescu & Dechter, 2009; Otten & Dechter, 2011).

### 2.2 CREDAL NETWORKS

A set of probability distributions for variable $X$ is called a *credal set* and is denoted by $K(X)$ (Levi, 1980). Similarly, a *conditional credal set* is a set of conditional distributions, obtained by applying Bayes rule to each distribution in a credal set of joint distributions (Walley, 1991). We consider credal sets that are closed and convex with a finite number of vertices. Two credal sets $K(X|Y = y_1)$ and

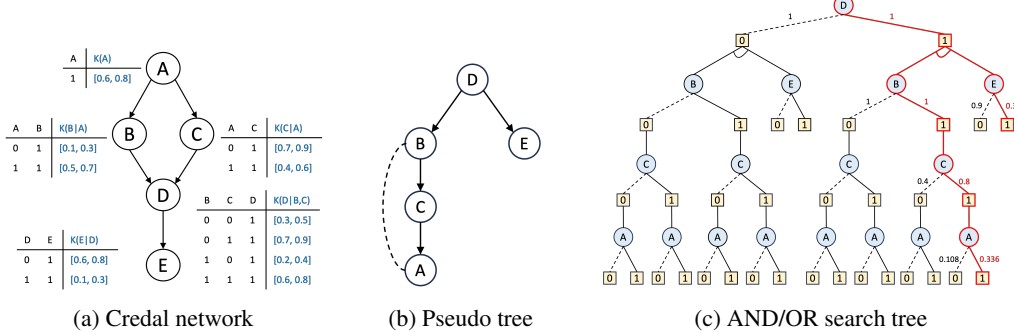

(a) Credal network    (b) Pseudo tree    (c) AND/OR search tree

Figure 1: Example of a credal network and its AND/OR search space.

$K(X|Y = y_2)$, where $y_1 \neq y_2$ are two values in variable $Y$'s domain, are called *separately specified* if there is no constraint on the first set that is based on the properties of the second set.

A *credal network* (CN) (Cozman, 2000) is defined by a tuple $\langle \mathbf{X}, \mathbf{D}, \mathbf{K}, G \rangle$, where $\mathbf{X} = \{X_1, \ldots, X_n\}$ is a set of discrete variables with finite domains $\mathbf{D} = \{D_1, \ldots, D_n\}$, $G$ is a directed acyclic graph (DAG) over $\mathbf{X}$ as nodes, and $\mathbf{K} = \{K(X_i|\Pi_i = \pi_{ik})\}$ is a set of separately specified conditional credal sets for each variable $X_i$ and each $k$-th configuration $\pi_{ik}$ of its parents $\Pi_i$ in $G$. The *strong extension* $K(\mathbf{X})$ of a credal network is the *convex hull* (denoted CH) of all joint distributions that satisfy the following Markov property: every variable is strongly independent of its non-descendants conditional on its parents (see Cozman (2000) for more details).

$$K(\mathbf{X}) = CH\{P(\mathbf{X}) \ : \ P(\mathbf{X}) = \prod_{i=1}^{n} P(X_i|\pi_{ik}), P(X_i|\pi_{ik}) \text{ is a vertex of } K(X_i|\Pi_i = \pi_{ik})\} \quad (2)$$

It can be shown that the strong extension $K(\mathbf{X})$ can be built from the extreme points of the conditional local credal sets denoted by $\text{ext}(K(X_i|\Pi_i = \pi_{ik}))$ (Mauá & Cozman, 2020).

**Example 1.** *Figure 1a shows a simple credal network with five bi-valued variables $\{A, B, C, D, E\}$. The local conditional credal sets are given by closed probability intervals. For example, we have that $0.1 \leq P(B = 1|A = 0) \leq 0.3$ and $0.5 \leq P(B = 1|A = 1) \leq 0.7$, respectively.*

In credal networks, there may be multiple distributions that admit maximal assignments (Mauá & Cozman, 2020). Therefore, we define the following *maximax* and *maximin* credal MAP tasks:

**Definition 1** (maximax MAP). *Given a credal network $\mathcal{C} = \langle \mathbf{X}, \mathbf{D}, \mathbf{K}, G \rangle$ and evidence $\mathbf{e}$ on $\mathbf{E} \subseteq \mathbf{X}$, the maximax MAP task is finding the assignment $\overline{\mathbf{y}}^*$ to $\mathbf{Y} = \mathbf{X} \setminus \mathbf{E}$ with maximum upper probability:*

$$\overline{\mathbf{y}}^* = \arg\max_{\mathbf{y} \in \Omega(\mathbf{Y})} \max_{P(\mathbf{Y}, \mathbf{e}) \in K(\mathbf{X})} \prod_{i=1}^{n} P(X_i|\Pi_i) \quad (3)$$

**Definition 2** (maximin MAP). *Given a credal network $\mathcal{C} = \langle \mathbf{X}, \mathbf{D}, \mathbf{K}, G \rangle$ and evidence $\mathbf{e}$ on $\mathbf{E} \subseteq \mathbf{X}$, the maximin MAP task is finding the assignment $\mathbf{y}^*$ to $\mathbf{Y} = \mathbf{X} \setminus \mathbf{E}$ with maximum lower probability:*

$$\underline{\mathbf{y}}^* = \arg\max_{\mathbf{y} \in \Omega(\mathbf{Y})} \min_{P(\mathbf{Y}, \mathbf{e}) \in K(\mathbf{X})} \prod_{i=1}^{n} P(X_i|\Pi_i) \quad (4)$$

It is easy to see that the upper probability (or value) of an assignment $\mathbf{x} = (x_1, \ldots, x_n)$ to $\mathbf{X}$ can be calculated as $\overline{P}(\mathbf{x}) = \prod_{i=1}^{n} \overline{P}(x_i|\pi_i)$, where $x_i$ and $\pi_i$ are $X_i$ and $\Pi_i$'s configurations in $\mathbf{x}$, and $\overline{P}(x_i|\pi_i) = \max \text{ext}(K(x_i|\pi_i))$ is the extreme point of $K(x_i|\pi_i)$ with the highest value. Similarly, the lower probability of $\mathbf{x}$ is $\underline{P}(\mathbf{x}) = \prod_{i=1}^{n} \underline{P}(x_i|\pi_i)$, where $\underline{P}(x_i|\pi_i) = \min \text{ext}(K(x_i|\pi_i))$ is the extreme point of $K(x_i|\pi_i)$ with the smallest value.

**Example 2.** *Consider again the credal network from Figure 1a and let $\mathbf{x} = (1, 1, 0, 0, 0)$ be a complete assignment to variables A, B, C, D and E. In this case, the conditional local credal sets $K(A = 1)$ and $K(B = 1|A = 1)$ have 2 unique extreme points each, i.e., $\text{ext}(K(A = 1)) = \{0.6, 0.8\}$ and $\text{ext}(K(B = 1|A = 1)) = \{0.5, 0.7\}$, respectively. The lower and upper probabilities of the assignment can be computed as $\underline{P}(\mathbf{x}) = 0.0144$ and $\overline{P}(\mathbf{x}) = 0.10752$, respectively.*

MAP inference in credal networks can also be shown to be NP-hard (Kwisthout, 2011; Campos & Cozman, 2005). Despite sharing the same complexity class as Bayesian MAP, credal MAP involves an optimization step over the extreme points of the joint credal set, making it significantly more challenging to solve in practice. However, unlike Bayesian MAP, currently there are no established algorithmic frameworks for *exact* MAP inference in credal networks.

## 3 BRANCH-AND-BOUND SEARCH FOR CREDAL MAP

We present now the first depth-first branch-and-bound search algorithms to exactly solve the *maximax* and *maximin* MAP tasks in credal networks. These algorithms explore an AND/OR representation of the search space which exploits the problem structure and has led to significant improvements in the search for MAP explanations in Bayesian networks (Marinescu & Dechter, 2009).

### 3.1 AND/OR SEARCH SPACES FOR CREDAL NETWORKS

The AND/OR search space which is defined relative to a *pseudo tree* capturing problem decomposition (Freuder & Quinn, 1985) has never been considered in the context of credal networks. Here, we extend and leverage it to facilitate the credal maximax and maximin MAP inference tasks.

**Definition 3** (pseudo tree). *A pseudo tree of an undirected graph $G = (V, E)$ is a directed rooted tree $T = (V, E')$ such that every arc of $G$ not included in $E'$ is a back-arc in $T$, namely, it connects a node in $T$ to one of its ancestors. The arcs in $E'$ may not all be included in $E$.*

Given a credal network $\langle \mathbf{X}, \mathbf{D}, \mathbf{K}, G \rangle$ and pseudo tree $T$ of $G$, the AND/OR search tree $S_T$ based on $T$ has alternating levels of OR nodes corresponding to the variables and AND nodes corresponding to the values of the OR parent's variable, with edges weighted according to the extreme point of the conditional local credal sets in $\mathbf{K}$. The size of the AND/OR search tree is bounded exponentially by the depth of the pseudo tree rather than the number of variables (Dechter & Mateescu, 2007).

A *solution tree* $\hat{\mathbf{x}}$ of $S_T$ is a subtree that: (1) contains the root of $S_T$ ; (2) if an internal OR node $n \in S_T$ is in $\hat{\mathbf{x}}$ then $n$ is labeled by a variable and exactly one of its children is in $\hat{\mathbf{x}}$; (3) if an internal AND node $n \in S_T$ is in $\hat{\mathbf{x}}$ then all its OR children labeled variables are in $\hat{\mathbf{x}}$.

Each edge from an OR node $X_i$ to its AND child $\langle X_i, x_i \rangle$ is associated with a weight $w(X_i, x_i)$. For *maximax* MAP, the weight is defined by the product of the upper probabilities corresponding to the extreme points of the conditional local credal sets $K(X_j | \pi_{jk})$ whose scopes mention variable $X_i$ and are completely instantiated along the path from the root of $S_T$ to $\langle X_i, x_i \rangle$. For *maximin* MAP, we consider the lower probabilities instead. Each node $n$ in $S_T$ is associated with a value $v(n)$ that captures the optimal maximax or maximin MAP value of the conditioned subproblem rooted at $n$. Clearly, $v(n)$ can be computed recursively based on the values of $n$'s successors and the corresponding edge weights: OR nodes by maximization and AND nodes by multiplication. The value of the optimal solution is therefore given by the value $v(s)$ of the root node $s$ of $S_T$.

**Example 3.** *Figure 1c we show the AND/OR search tree of the credal network from Figure 1a relative to the pseudo tree given in Figure 1b. The solution tree $\hat{\mathbf{x}}$ corresponding to the assignment $(A = 1, B = 1, C = 1, D = 1, E = 1)$ is highlighted, and its maximax MAP value, for example, is obtained by multiplying the weights associated with the OR-to-AND edges in $\hat{\mathbf{x}}$. In this case, the weight $w(A, 1)$ of the edge from $A$ to $\langle A, 1 \rangle$ in $\hat{\mathbf{x}}$ is $w(A, 1) = \overline{P}(A = 1) \cdot \overline{P}(B = 1 | A = 1) \cdot \overline{P}(C = 1 | A = 1) = 0.336$, where $\overline{P}(A = 1) = \max ext(K(A = 1)) = 0.8$, $\overline{P}(B = 1 | A = 1) = \max ext(K(B = 1 | A = 1)) = 0.7$ and $\overline{P}(C = 1 | A = 1) = \max ext(K(C = 1 | A = 1)) = 0.6$, respectively.*

### 3.2 AND/OR BRANCH-AND-BOUND SEARCH FOR CREDAL MAP

We present an AND/OR Branch and Bound algorithm designed to solve the maximax and maximin MAP tasks. This algorithm builds upon recent AND/OR search schemes developed for MAP inference in Bayesian networks (Marinescu & Dechter, 2009), extending them to credal networks.

Algorithm 1 outlines the AND/OR Branch and Bound (AOBB) approach for solving the maximax MAP problem in credal networks. We denote the current partial solution, the evidence and the value of the best solution found so far as $\hat{\mathbf{x}}$, $\mathbf{e}$, and $S$ respectively. The algorithm assumes that variables

---

**Algorithm 1** AND/OR Branch-and-Bound Search for *Maximax/Maximin* Credal MAP

---

1: **procedure** AOBB($\mathcal{C} = \langle \mathbf{X}, \mathbf{D}, \mathbf{K}, \rangle, \mathbf{e}, T$)
2:   **if** $\mathbf{X} = \emptyset$ **then**
3:     **return** 1
4:   **else**
5:     $X_k \leftarrow$ SELECTVAR($\mathbf{X}$) according to $T$
6:     **if** $X_k$ is evidence variable **then**
7:       $D_k = \{x_k\}$ such that $x_k \in \mathbf{e}$
8:     Initialize $v(X_k) \leftarrow 0$
9:     **for all** values $x_k \in D_k$ **do**
10:       $\hat{\mathbf{x}}_k \leftarrow \hat{\mathbf{x}}_k \cup \{X_k = x_k\}$
11:       Evaluate $f(\hat{\mathbf{x}}_k)$
12:       **if** $f(\mathbf{x}_k) > S$ **then**

13:         Initialize $v(X_k, x_k) \leftarrow 1$
14:         **for all** children $X_q$ of $X_k$ in $T$ **do**
15:         $val \leftarrow$ AOBB($\mathcal{C}_q, \mathbf{e}, T$)
16:         $v(X_k, x_k) \leftarrow v(X_k, x_k) \cdot val$
17:       **else**
18:         Set $v(X_k, x_k) \leftarrow 0$
19:       $\hat{\mathbf{x}}_k \leftarrow \hat{\mathbf{x}}_k \setminus \{X_k = x_k\}$
20:       $val \leftarrow w(X_k, x_k) \cdot v(X_k, v_k)$
21:       Update $v(X_k) \leftarrow \max(v(X_k), val)$
22:     **if** $X_k$ is root of $T$ **then**
23:       Update $S \leftarrow \max(S, v(X_k))$
24:   **return** $v(X_k)$

---

are selected statically based on a pseudo tree $T$. A heuristic evaluation function, $f(\hat{\mathbf{x}})$, computes an upper bound on the optimal maximax MAP extension of $\hat{\mathbf{x}}$. For the maximin MAP task, only the computation of edge weights and the heuristic evaluation function needs to be adjusted.

If the set $\mathbf{X}$ is empty, the result is trivially computed (line 3). Otherwise, AOBB selects the next variable $X_k$ in $T$ and iterates over its domain values (i.e., its AND successors) to compute the OR node value $v(X_k)$ (lines 8-21). Subsequently, the algorithm attempts to prune unpromising domain values by comparing the upper bound $f(\hat{\mathbf{x}})$ of the current partial solution tree $\bar{\mathbf{x}}$ to the value $S$ of the current best solution tree found which is maintained by the root node $s$ of the search space (line 12). For each domain value $x_k$ of $X_k$, the problem rooted by the AND node labeled $\langle X_k, x_k \rangle$ is decomposed into $r$ independent subproblems $\mathcal{C}_q = \langle \mathbf{X}_q, \mathbf{D}_q, \mathbf{K}_q \rangle$, one for each child $X_q$ of $X_k$ in $T$. Note that if $X_k$ is an evidence variable then its domain is just $D_k = \{x_k\}$ where $x_k \in \mathbf{e}$ (lines 6-7). These problems are then solved independently and their results are accumulated by the AND node value $v(X_k, x_k)$ (lines 14–16). After trying all possible values of variable $X_k$, the maximax MAP value of the subproblem rooted by $X_k$ is $v(X_k)$ and is returned (line 24). Finally, the optimal maximax MAP value for the original problem is returned by the root node $s$ of the search space.

AOBB computes its guided heuristic function $f(\hat{\mathbf{x}})$ using an improved mini-bucket based bounding scheme which we will describe in detail in Section 4. The heuristic can be pre-compiled along the reverse order of a depth-first traversal of the pseudo tree (which corresponds to an elimination order).

**Theorem 1** (complexity). *Given a credal network $\mathcal{C} = \langle \mathbf{X}, \mathbf{D}, \mathbf{K}, G \rangle$ and evidence $\mathbf{e}$, the time and space complexities of algorithm AOBB are $O(n \cdot d^h)$ and $O(n)$, respectively, where $h$ is the depth of the pseudo tree $T$ of $G$, $n$ is the number of variables and $d$ bounds their domain sizes.*

## 4   MINI-BUCKETS FOR CREDAL MAP

We next describe novel partitioning-based bounds (aka mini-bucket bounds) that are compatible with AOBB search for both maximax and maximin MAP. Although the mini-bucket bounds have proven effective in guiding search algorithms for MAP in Bayesian networks (Kask & Dechter, 2001; Marinescu & Dechter, 2009), they have not yet been explored in the context of credal networks.

### 4.1   POTENTIALS AND THEIR APPROXIMATIONS

Unlike in Bayesian networks, variable elimination schemes for the credal MAP tasks must operate on sets of probability functions called *potentials* (Mauá & Cozman, 2020; Marinescu et al., 2023):

**Definition 4** (potential). *Given a set of variables $\mathbf{Y}$, a potential $\phi(\mathbf{Y})$ is a set of non-negative real-valued functions $p(\mathbf{Y})$ on $\mathbf{Y}$. The product of two potentials $\phi(\mathbf{Y})$ and $\psi(\mathbf{Z})$ is $\phi(\mathbf{Y}) \cdot \psi(\mathbf{Z}) = \{p \cdot q : p \in \phi(\mathbf{Y}), q \in \psi(\mathbf{Z})\}$. The max-marginal $\max_{\mathbf{Z}} \phi(\mathbf{Y})$ of a potential $\phi(\mathbf{Y})$ with respect to a subset of variables $\mathbf{Z} \subseteq \mathbf{Y}$ is defined by $\max_{\mathbf{Z}} \phi(\mathbf{Y}) = \{\max_{\mathbf{Z}} p(\mathbf{Y}) : p \in \phi(\mathbf{Y})\}$.*

---

**Algorithm 2** Mini-Buckets with Moment-Matching for Maximax MAP

1: **procedure** MBMM($\mathcal{C} = \langle \mathbf{X}, \mathbf{D}, \mathbf{K} \rangle, i, M$)
2:   Initialize $\Gamma \leftarrow \emptyset$
3:   **for all** variable $X_k \in \mathbf{X}$ **do**
4:     Let $\phi_k = \{p : p \in ext(K(X_k|\Pi_k))\}$
5:     Update $\Gamma = \Gamma \cup \{\text{PLUB}(\phi_k, M)\}$
6:   Create elimination ordering $o : X_1, \ldots, X_n$
7:   **for all** variable $X_k \in o$ **do**
8:     ▷ Create bucket $\Gamma_k$ and mini-buckets $Q_{kr}$
9:     Let $\Gamma_k = \{\phi : \phi \in \Gamma, X_k \in vars(\phi)\}$
10:    Update $\Gamma = \Gamma \setminus \Gamma_k$
11:      ▷ Create mini-buckets $Q_{k1}, \ldots, Q_{kR}$
12:    Partition $\Gamma_k$ into $\{Q_{k1}, \ldots, Q_{kR}\}$
13:    **for all** $r = 1$ to $R$ **do**
14:     Let $\phi_{kr} = \text{PLUB}(\prod_{\phi \in Q_{kr}} \phi, M)$
15:     Let $\mathbf{Y}_k = vars(Q_{kr}) \setminus X_k$
16:       ▷ Moment-matching on max marginals
17:    Let $\mu_r = \text{PLUB}(\max_{\mathbf{Y}_r} \phi_{kr}, 1)$
18:    Let $\mu = \left(\prod_r \mu_r\right)^{1/R}$
19:    Update $\phi_{kr} = \phi_{kr} \cdot \left(\frac{\mu}{\mu_r}\right)$
20:      ▷ Compute the downward messages
21:    **for all** $r = 1$ to $R$ **do**
22:     Let $\lambda_r^k = \text{PLUB}(\max_{X_k} \phi_{kr}, M)$
23:     Update $\Gamma = \Gamma \cup \{\lambda_r^k\}$
24:   **return** $\max(\prod_{\phi \in \Gamma} \phi)$

---

A non-negative probability function $p(\mathbf{Y})$ defined over variables $\mathbf{Y}$ can be viewed as a vector in $\mathbb{R}^m$, where $\Omega(\mathbf{Y})$ is the Cartesian product of the domains of the variables in $\mathbf{Y}$, and $m = |\Omega(\mathbf{Y})|$ is its cardinality. We say that $p(\mathbf{Y}) \leq q(\mathbf{Y})$ if and only if $\forall \mathbf{y} \in \Omega(\mathbf{Y}), p(\mathbf{y}) \leq q(\mathbf{y})$. Clearly, $\leq$ is a partial order. Therefore, a pruning operator $\max \phi(\mathbf{Y})$ that selects the maximal elements of a potential $\phi(\mathbf{Y})$ is defined relative to $\leq$ as: $\max \phi(\mathbf{Y}) = \{p(\mathbf{Y}) \in \phi(\mathbf{Y}) : \nexists q(\mathbf{Y}) \in \phi(\mathbf{Y}), p(\mathbf{Y}) \leq q(\mathbf{Y})\}$.

Furthermore, since the multiplication operator can significantly increase the size of potentials, we require a potential to have a restricted cardinality, at most $M (\geq 1)$. Therefore, we need an operator that takes the potential $\phi(\mathbf{Y})$, with $|\phi(\mathbf{Y})| > M$, and reduces it to a smaller potential $\phi'(\mathbf{Y})$ with cardinality at most $M$, while ensuring that $\phi'(\mathbf{Y})$ provides an upper bound on $\phi(\mathbf{Y})$. Specifically, for every $p(\mathbf{Y}) \in \phi(\mathbf{Y})$ there exists $q(\mathbf{Y}) \in \phi'(\mathbf{Y})$ such that $p(\mathbf{Y}) \leq q(\mathbf{Y})$. To achieve this, we utilize the Pareto Least Upper Bound (PLUB) of vectors in $\mathbb{R}^m$, defined as follows:

**Definition 5** (PLUB). *The* Pareto Least Upper Bound (PLUB) $\vec{v} \in \mathbb{R}^m$ *of a set of* $k$ *vectors* $\{\vec{v}_1, ..., \vec{v}_k\} \in \mathbb{R}^m$ *is given by* $\vec{v} = \max_{j=1}^k \vec{v}_j$, *where the* max *is applied point-wise.*

A simple procedure to compute the upper bound $\phi'(\mathbf{Y})$ of $\phi(\mathbf{Y})$ is to group the elements of $\phi(\mathbf{Y})$ into $M$ clusters based on minimizing the Manhattan distance to each cluster's centroid (i.e., minimize $\sum_{i=1}^m |p_i - r_i|$, where $p_i$ and $r_i$ are the $i$-th components of $p$ and $r$, respectively). Then, for each cluster we replace its components with their Pareto least upper bound.

## 4.2 THE MAXIMAX MAP CASE

Algorithm 4 adapts the mini-bucket approximation scheme developed for Bayesian MAP (Dechter & Rish, 2003) to the maximax MAP task in credal networks. Specifically, the MBMM($i$) algorithm partitions large buckets into smaller subsets, called *mini-buckets*, each containing at most $i$ distinct variables (aka the $i$-bound). The mini-buckets are processed separately by maximizing out the bucket variable from the combination of potentials within each mini-bucket. Furthermore, the algorithm avoids generating prohibitively large potentials at each elimination step by approximating both the intermediate and the original potentials with their Pareto least upper bounds of size $M$.

While the PLUB-based approximation of potentials may result in a looser overall upper bound, this bound can be tightened further using a moment-matching re-parameterization scheme inspired by (Ihler et al., 2012). Consider the following simple example with three variables $A, B, C$ and two binary potentials $\phi(A, B)$ and $\phi(A, C)$. In this case, we can rewrite the mini-bucket upper bound as:

$$\max_A \left[\phi(A, B) \cdot \phi(A, C)\right] = \max_A \left[\phi(A, B) \cdot \lambda_1(A) \cdot \phi(A, C) \cdot \lambda_2(A)\right]$$

$$\leq \max_A \left[\phi(A, B) \cdot \lambda_1(A)\right] \cdot \max_A \left[\phi(A, C) \cdot \lambda_2(A)\right]$$

where $\lambda_1(A)$ and $\lambda_2(A)$ are two auxiliary positive functions such that $\lambda_1(A) \cdot \lambda_2(A) = 1$. A simple choice for the $\lambda$ functions is to use the max-marginals on $A$. Let $\varphi_1(A) = \max_B \phi(A, B)$ and $\varphi_2(A) = \max_C \phi(A, C)$ be the max-marginal potentials on $A$, and let $\mu_1(A)$ and $\mu_2(A)$ be their

Table 1: Quality of Heuristics for Maximax MAP on 100 variables `random` networks. Average CPU time (sec) and number of nodes expanded using $i$-bounds from 2 to 10. Time limit 1 hour.

| size | algorithm | $i = 2$ | | $i = 4$ | | $i = 6$ | | $i = 8$ | | $i = 10$ | |
|------|-----------|------|-------|------|-------|--------|-------|---------|-------|---------|-------|
| | | time | nodes | time | nodes | time | nodes | time | nodes | time | nodes |
| | AOBB+MB($i$,1) | 2.90 | 29603 | 0.99 | 13934 | 0.54 | 6877 | 0.25 | 5280 | 0.18 | 2352 |
| | AOBB+MB($i$,10) | 3.06 | 29603 | 1.47 | 13934 | 18.65 | 6877 | 501.36 | 5834 | 809.46 | 2582 |
| 100 | AOBB+MB($i$,50) | 3.25 | 29603 | 2.07 | 13934 | 558.29 | 7609 | 2781.15 | 2492 | 3284.57 | 251 |
| | AOBB+MBMM($i$,1) | 2.25 | 21057 | **0.39** | 8316 | **0.40** | 4448 | **0.15** | 3544 | **0.20** | 1807 |
| | AOBB+MBMM($i$,10) | **1.97** | 21057 | 0.75 | 8316 | 15.99 | 4448 | 488.84 | 3906 | 797.66 | 1978 |
| | AOBB+MBMM($i$,50) | 2.22 | 21057 | 1.04 | 8316 | 548.81 | 4910 | 2800.54 | 1633 | 3312.94 | 108 |
| | AOBB+MB($i$) | 2.87 | 29603 | 2.77 | 13934 | 972.37 | 7336 | - | - | - | - |

PLUB approximations of size 1. If $\mu(A) = \sqrt{\mu_1(A) \cdot \mu_2(A)}$ is their geometric mean, then for our re-parameterization we can use: $\lambda_1(A) = \frac{\mu(A)}{\mu_1(A)}$ and $\lambda_2(A) = \frac{\mu(A)}{\mu_2(A)}$, respectively. We have that:

**Theorem 2** (complexity). *Algorithm MBMM($i$) computes an upper bound on the optimal maximax MAP value. The time and space complexity is $O(n \cdot M^2 \cdot d^i)$, where $i$ is the $i$-bound, $n$ is the number of variables, $d$ bounds the domain sizes and $M$ bounds the cardinality of the potentials.*

### 4.3 THE MAXIMIN MAP CASE

For the maximin MAP task, we define the pruning operator $\min \phi(\mathbf{Y})$ to identify the minimal elements of a potential $\phi(\mathbf{Y})$ according to the same partial order $\leq$ used in the maximax MAP scenario, as follows: $\min(\phi(\mathbf{Y})) = \{p(\mathbf{Y}) \in \phi(\mathbf{Y}) : \nexists q(\mathbf{Y}) \in \phi(\mathbf{Y}), q(\mathbf{Y}) \leq p(\mathbf{Y})\}\}$.

However, the $\max$ and $\min$ operators in Equation 4 do not commute. As a result, the variable elimination scheme that uses the $\min$ pruning operator is not exact anymore and only yields an upper bound on the optimal maximin MAP value. Even when the mini-bucket approximation is enhanced with cost-shifting via moment matching, it continues to provide an upper bound – though these are generally much looser than those obtained in the maximax MAP setting. Our experimental results clearly demonstrate that the mini-bucket bounds for maximin MAP are substantially weaker, and the associated search algorithms face significant challenges as a result.

## 5 EXPERIMENTS

We evaluate the proposed branch-and-bound search algorithms for maximax and maximin MAP on random credal networks and credal networks derived from real-world applications. All competing algorithms were implemented in C++ and the experiments were run on a machine with a 16-core 3GHz CPU and 128GB of RAM running Ubuntu Linux 24.04.

### 5.1 ALGORITHMS, BENCHMARKS AND MEASURES OF PERFORMANCE

Our proposed AND/OR Branch and Bound (AOBB) algorithm is equipped with the following versions of the mini-bucket heuristics: (1) mini-buckets without potential approximation and moment-matching denoted by AOBB+MB($i$), (2) mini-buckets with potential approximation of size $M$ only, denoted by AOBB+MB($i$, $M$), and (3) mini-buckets with both potential approximation and moment-matching, denoted by AOBB+MBMM($i$, $M$). For comparison, we also ran the OR Branch and Bound (BB) counterparts guided by the same heuristic schemes, denoted by BB+MB($i$), BB+MB($i$, $M$), and BB+MBMM($i$, $M$). Unlike the former methods, the latter ones are not sensitive to the underlying problem structure. For reference, we also ran the brute-force depth-first search denoted by DFS that exhaustively enumerates all possible MAP assignments (see Appendix).

For our purpose, we generate `random` and $m$-by-$m$ `grid` credal networks. Specifically, for random networks, we vary the number of variables $n \in \{100, 150, 200\}$ and, for grids, we choose $m \in \{10, 14, 16\}$, respectively. For each problem size, we generate 10 random problem instances. In all cases, the maximum domain size is set to 2 and the local conditional credal sets are generated uniformly at random as probability intervals. In addition, we consider a set of 15 credal networks

Table 2: Results for Maximax MAP on `random` and `grid` credal networks. Average CPU time (sec) and number of nodes expanded using mini-bucket $i$-bounds from 2 to 10. Time limit 1 hour.

| size (w,h) | algorithm | $i=2$ | | $i=4$ | | $i=6$ | | $i=8$ | | $i=10$ | |
|---|---|---|---|---|---|---|---|---|---|---|---|
| | | time | nodes | time | nodes | time | nodes | time | nodes | time | nodes |
| | | | | | | `random` credal networks | | | | | |
| 100 (18,28) | BB+MB(i) | 3525.20 | 40136432 | 1014.36 | 6399801 | 1789.33 | 123229 | - | - | - | - |
| | BB+MBMM(i,1) | 2481.97 | 34115031 | 154.77 | 1717539 | 29.81 | 362314 | 1.00 | 33382 | 0.86 | 15106 |
| | AOBB+MB(i) | 2.87 | 29603 | 2.77 | 13934 | 972.37 | 7336 | - | - | - | - |
| | AOBB+MBMM(i,1) | **2.25** | 21057 | **0.39** | 8316 | **0.40** | 4448 | **0.15** | 3544 | **0.20** | 1807 |
| 150 (27,38) | BB+MB(i) | - | - | - | - | - | - | - | - | - | - |
| | BB+MBMM(i,1) | - | - | 3253.52 | 35114767 | 1119.96 | 15193688 | 1046.74 | 16149075 | 389.39 | 4938087 |
| | AOBB+MB(i) | 156.22 | 1452909 | 77.77 | 492383 | 1275.38 | 419109 | 3252.94 | 411225 | - | - |
| | AOBB+MBMM(i,1) | **68.48** | 612721 | **23.95** | 323892 | **11.55** | 158230 | **6.56** | 148781 | **8.43** | 133786 |
| 200 (36,48) | BB+MB(i) | - | - | - | - | - | - | - | - | - | - |
| | BB+MBMM(i,1) | - | - | - | - | - | - | 2780.10 | 41392520 | 2342.75 | 37828931 |
| | AOBB+MB(i) | 1555.37 | 8510209 | 1126.37 | 4969424 | 2285.98 | 813793 | 3458.96 | 96389 | - | - |
| | AOBB+MBMM(i,1) | **1155.54** | 8448489 | **1108.73** | 7985450 | **197.49** | 2193611 | **224.53** | 1974790 | **364.89** | 2768420 |
| | | | | | | `grid` credal networks | | | | | |
| 100 (14,38) | BB+MB(i) | - | - | - | - | - | - | - | - | - | - |
| | BB+MBMM(i,1) | - | - | 3287.93 | 45186026 | 362.59 | 5259684 | 0.07 | 699 | 0.13 | 123 |
| | AOBB+MB(i) | 3.82 | 65648 | 0.23 | 2138 | - | - | - | - | - | - |
| | AOBB+MBMM(i,1) | **0.74** | 19531 | **0.30** | 2138 | **0.06** | 1235 | **0.07** | 226 | **0.21** | 107 |
| 144 (18,49) | BB+MB(i) | - | - | - | - | - | - | - | - | - | - |
| | BB+MBMM(i,1) | - | - | - | - | 3244.43 | 38700863 | 38.68 | 736998 | 0.35 | 1810 |
| | AOBB+MB(i) | 33.80 | 524460 | 0.42 | 8355 | - | - | - | - | - | - |
| | AOBB+MBMM(i,1) | **18.45** | 247311 | **0.73** | 8325 | **0.27** | 7413 | **0.11** | 1046 | **0.32** | 327 |
| 196 (20,57) | BB+MB(i) | - | - | - | - | - | - | - | - | - | - |
| | BB+MBMM(i,1) | - | - | - | - | - | - | 3240.10 | 39599609 | 1072.23 | 16685172 |
| | AOBB+MB(i) | 961.58 | 10746465 | 1.54 | 32950 | 3478.82 | 32950 | - | - | - | - |
| | AOBB+MBMM(i,1) | **395.68** | 4201397 | **2.19** | 32950 | **1.03** | 31339 | **1.18** | 24508 | **1.83** | 32948 |

Table 3: Results for Maximax MAP on real-world credal networks. CPU time (sec) and number of nodes expanded using mini-bucket $i$-bounds from 2 to 10. Time limit 1 hour.

| instance (n, w, h) | algorithm | $i=2$ | | $i=4$ | | $i=6$ | | $i=8$ | | $i=10$ | |
|---|---|---|---|---|---|---|---|---|---|---|---|
| | | time | nodes | time | nodes | time | nodes | time | nodes | time | nodes |
| alarm (37,4,12) | BB+MB($i$) | 2030.54 | 150170 | 3.98 | 50 | 5.67 | 39 | 6.07 | 39 | 3.87 | 39 |
| | BB+MBMM($i$,1) | 5.52 | 4535 | 3.77 | 39 | 3.99 | 39 | 4.92 | 39 | 5.36 | 39 |
| | AOBB+MB($i$) | 5.78 | 85 | 4.81 | 42 | 5.25 | 39 | 7.64 | 39 | 6.60 | 39 |
| | AOBB+MBMM($i$,1) | **2.62** | 52 | **2.82** | 39 | **2.80** | 39 | **5.38** | 39 | **2.80** | 39 |
| link (724,15,43) | BB+MB($i$) | - | - | - | - | - | - | - | - | - | - |
| | BB+MBMM($i$,1) | - | - | - | - | - | - | - | - | - | - |
| | AOBB+MB($i$) | 23.09 | 67424 | 3.74 | 1772 | - | - | - | - | - | - |
| | AOBB+MBMM($i$,1) | **9.77** | 33603 | **2.97** | 1004 | **2.99** | 978 | **2.79** | 978 | **2.58** | 793 |
| mastermind1 (1220,20,56) | BB+MB($i$) | - | - | - | - | - | - | - | - | - | - |
| | BB+MBMM($i$,1) | - | - | - | - | - | - | - | - | - | - |
| | AOBB+MB($i$) | - | - | 102.60 | 34669 | - | - | - | - | - | - |
| | AOBB+MBMM($i$,1) | - | - | **26.64** | 34493 | **9.96** | 17619 | **9.97** | 17619 | **9.83** | 17619 |
| mastermind3 (3692,39,92) | BB+MB($i$) | - | - | - | - | - | - | - | - | - | - |
| | BB+MBMM($i$,1) | - | - | - | - | - | - | - | - | - | - |
| | AOBB+MB($i$) | - | - | - | - | - | - | - | - | - | - |
| | AOBB+MBMM($i$,1) | - | - | - | - | **3264.92** | 2180932 | **3088.55** | 2172466 | **3036.91** | 2167200 |

derived from real-world Bayesian networks[1] by converting the probability values in the conditional probability tables into probability intervals. For all our problem instances, we ensure that the difference between the lower and upper bounds of the probability intervals was at most 0.3. The problem sizes were deliberately chosen to ensure they could be solved exactly within the specified time limit. Finally, we experiment with maximax and maximin MAP tasks with no evidence.

In all of our experiments, we report the CPU time in seconds and the number of nodes expanded during the search. We also record the number of variables ($n$), the induced width ($w$) and the height of the pseudo trees ($h$) for all of our benchmarks. The best performance points are highlighted. All competing algorithms were allocated a 1 hour time limit and 10GB of memory. The "-" symbol indicates that the respective algorithm exceeded its time or memory budget.

## 5.2 QUALITY OF HEURISTICS

Table 1 shows the average CPU time in seconds and number of nodes expanded by AOBB when guided by the MB($i$), MB($i$, $M$) and MBMM($i$, $M$) heuristics for solving maximax MAP on random

---

[1]Available at https://www.bnlearn.com/bnrepository/

Table 4: Results for Maximin MAP on real-world credal networks. CPU time (sec) and number of nodes expanded using mini-bucket $i$-bounds from 2 to 12. Time limit 1 hour.

| instance $(n, w, h)$ | algorithm | $i = 2$ | | $i = 4$ | | $i = 6$ | | $i = 8$ | | $i = 10$ | | $i = 12$ | |
|---|---|---|---|---|---|---|---|---|---|---|---|---|---|
| | | time | nodes | time | nodes | time | nodes | time | nodes | time | nodes | time | nodes |
| alarm (37,4,12) | BB+MB($i$) | 128.78 | 2522160 | 110.17 | 2522160 | 96.03 | 2522160 | 98.37 | 2522160 | 77.19 | 2522160 | 81.93 | 2522160 |
| | BB+MBMM($i$,1) | 126.35 | 2522160 | 106.60 | 2522160 | 101.46 | 2522160 | 97.39 | 2522160 | 89.28 | 2522160 | 63.16 | 2522160 |
| | AOBB+MB($i$) | **6.90** | 460 | 3.86 | 348 | **6.79** | 348 | **5.60** | 348 | **5.68** | 348 | **5.49** | 348 |
| | AOBB+MBMM($i$,1) | 6.96 | 394 | **3.77** | 348 | 7.11 | 348 | 6.64 | 348 | 6.17 | 348 | 6.91 | 348 |
| link (724,15,43) | BB+MB($i$) | - | - | - | - | - | - | - | - | - | - | - | - |
| | BB+MBMM($i$,1) | - | - | - | - | - | - | - | - | - | - | - | - |
| | AOBB+MB($i$) | - | - | 1663.40 | 7820555 | **1245.14** | 7448824 | **1180.95** | 7427647 | 1139.88 | 7405547 | 1072.66 | 7405953 |
| | AOBB+MBMM($i$,1) | - | - | **1538.40** | 7455392 | 1245.57 | 7406117 | 1188.28 | 7386176 | **1122.29** | 7383818 | **995.99** | 6862912 |
| mastermind1 (1220,20,56) | BB+MB($i$) | - | - | - | - | - | - | - | - | - | - | - | - |
| | BB+MBMM($i$,1) | - | - | - | - | - | - | - | - | - | - | - | - |
| | AOBB+MB($i$) | 49.00 | 64081 | **49.14** | 64240 | 34.42 | 62281 | 32.35 | 61822 | **28.90** | 61189 | 30.69 | 60236 |
| | AOBB+MBMM($i$,1) | **47.79** | 64081 | 49.17 | 64259 | **34.04** | 62281 | **32.14** | 61737 | 32.29 | 61275 | **30.17** | 61245 |
| mastermind3 (3692,39,92) | BB+MB($i$) | - | - | - | - | - | - | - | - | - | - | - | - |
| | BB+MBMM($i$,1) | - | - | - | - | - | - | - | - | - | - | - | - |
| | AOBB+MB($i$) | - | - | - | - | **1200.00** | 1357913 | 1099.27 | 1357053 | 1092.61 | 1362794 | 1103.99 | 1370757 |
| | AOBB+MBMM($i$,1) | - | - | - | - | 1205.83 | 1358178 | **1083.09** | 1360571 | **1077.06** | 1365064 | **1049.79** | 1367606 |

credal networks with 100 variables. The columns are indexed by the $i$-bound, and we varied $M$ between 1 and 50, respectively. We can see that all of the mini-bucket heuristics are competitive for the smallest $i$-bounds and all values of $M$ because the intermediate potentials do not grow too large in this case and, therefore, the computational overhead is reduced. However, as the $i$-bound and $M$ value increase, the size of intermediate potentials grows significantly due to much larger scope sizes, which eventually translates into increased overhead. We notice that using $M = 1$ yields the most cost-effective heuristics, especially for larger $i$-bounds which produce more accurate bounds that prune the search space very effectively (Marinescu & Dechter, 2009). The moment-matching cost-shifting scheme further tightens the heuristics, almost always leading to time savings, as previously observed for Bayesian networks (Ihler et al., 2012; Marinescu et al., 2014).

## 5.3 Results for Maximax and Maximin MAP

Table 2 summarizes the results obtained on the `random` and `grid` credal networks with the search algorithms guided by the MB($i$) and MBMM($i$, $M = 1$) heuristics. As before, algorithm AOBB+MB($i$) is competitive only at the smallest $i$-bounds due to the computational overhead associated with the larger intermediate potentials that are generated at larger $i$-bounds. Furthermore, the AND/OR search algorithms that exploit the problem structure and are guided by the MBMM($i$, $M = 1$) heuristics improve dramatically over their OR search counterparts, in some cases by up to 5 orders of magnitude, especially at relatively smaller $i$-bounds (e.g., $i = 4$ on 10-by-10 grids). As the $i$-bound increases, the corresponding heuristics tend to be more accurate, and this often translates into additional time savings for the AOBB+MBMM($i$, $M = 1$) algorithm. However, when the $i$-bound increases even further, the running time of AOBB+MBMM($i$, $M = 1$) starts to increase slightly because of the overhead associated with compiling the heuristics. The brute-force DFS algorithm could only solve problems with up to 20 variables, and therefore is omitted. These results are consistent with those obtained previously on Bayesian MAP (Marinescu & Dechter, 2009).

Table 3 reports the CPU time in seconds and number of nodes expanded on 4 real-world credal networks. The results show a similar pattern as before where the AND/OR search algorithm equipped with mini-buckets using moment-matching and potential approximation of size 1 outperforms dramatically its competitors, at all reported $i$-bounds. Furthermore, AOBB+MBMM($i$, $M = 1$) is the only algorithm that scales to problems with more than 3000 variables (e.g., `mastermind3`) and proves the optimality of the solutions obtained.

Table 4 shows the results for maximin MAP on real-world networks. As with maximax MAP, AND/OR search algorithms consistently outperform their OR counterparts across all $i$-bounds. However, maximin MAP proves significantly more challenging, primarily due to weaker heuristics that lead to larger search spaces and reduced performance (see Appendix for additional results).

## 5.4 Exact versus Local Search

In Table 5 we report the average CPU time obtained with the recent local search algorithms from Marinescu et al. (2023) which we adapted to maximax MAP. Specifically, we ran each of the Stochastic Local Search (SLS), Taboo Search (TS), Simulated Annealing (SA) and Guided Local

Table 5: Average CPU time in seconds for exact vs local search algorithms. Time limit 1 hour.

| size | AOBB+MBMM($i$,1) | SLS | TS | SA | GLS |
|---|---|---|---|---|---|
| | random credal networks | | | | |
| 20 | **0.00** | 49.60 | 46.34 | 33.61 | 55.83 |
| 50 | **0.01** | 184.46 | 107.5 | 98.24 | 175.69 |
| 100 | **0.10** | 372.75 | 188.92 | 196.96 | 352.78 |
| 150 | **2.62** | 565.95 | 223.03 | 300.46 | 529.20 |
| 200 | **109.25** | 681.60 | 438.32 | 326.83 | 563.61 |
| | grid credal networks | | | | |
| 25 | **0.01** | 53.21 | 44.77 | 38.20 | 59.95 |
| 49 | **0.01** | 186.56 | 68.27 | 59.30 | 169.44 |
| 100 | **0.05** | 350.69 | 171.27 | 164.97 | 327.31 |
| 144 | **0.09** | 421.20 | 203.42 | 207.58 | 424.90 |
| 196 | **0.14** | 572.15 | 312.78 | 362.54 | 456.23 |

Search (GLS) algorithms for a total of 10 iterations (i.e., random restarts) with a maximum of 100K flips per iteration. The random flip probability was set to 0.1, the taboo list had a maximum size of 1000, while the alpha and initial temperature used by SA were set to 0.1 and 100, respectively. We can see clearly that in this case the exact algorithm AOBB+MBMM($i$, 1) dominates the other competitors while proving the optimality of the solutions obtained.

While this paper primarily focuses on proving solution optimality, we note that our search schemes can be readily extended to efficient anytime algorithms, following the approach in Otten & Dechter (2011), to provide the best solution found so far at any point during the search. Furthermore, since the optimal maximax/maximin MAP assignment may not be unique, the proposed AND/OR algorithms can be equipped with a book-keeping mechanism similar to the one developed for the $k$-best MAP task in Bayesian networks (Dechter et al., 2012) to enable the enumeration of all optimal assignments.

## 6 RELATED WORK

Bayesian MAP has been extensively investigated over the years and several exact and approximate algorithmic frameworks have been developed such as stochastic local search (Kask & Dechter, 1999; Park, 2002; Hutter et al., 2005), variational approximation and message-passing schemes (Pearl, 1988; Dechter et al., 2002; Dechter & Rish, 2003; Wainwright et al., 2005; Kolmogorov, 2006; Ihler et al., 2012), or heuristic search (Kask & Dechter, 1999; Larrosa & Schiex, 2003; Marinescu & Dechter, 2009; Otten & Dechter, 2011). More recently, neural network based approximate solvers without solution guarantees have also been proposed (Arya et al., 2024; 2025). Credal MAP has received limited attention with some prior work on MAP inference in specialized models such as hidden Markov models with set-valued parameters (Mauá et al., 2016) and the approximate solvers for credal Marginal MAP developed recently by Marinescu et al. (2023). In contrast, our contribution addresses *exact* credal MAP inference with guarantees in general high-dimensional credal networks.

## 7 CONCLUSION

This paper significantly advances the field of MAP inference in credal networks by introducing novel depth-first branch-and-bound search algorithms. These algorithms leverage the AND/OR search space to effectively exploit the problem structure, and are further enhanced with a partitioning-based heuristic that combines potential approximations with cost-shifting strategies. Our empirical evaluations demonstrate that these new methods not only improve computational efficiency but also scale to large problems with over 3000 variables while guaranteeing optimality of solutions. Thus, our proposed approach addresses critical limitations of the state-of-the-art, providing robust and efficient solutions for MAP inference tasks in credal networks. Potential future directions include improving the mini-bucket heuristics for maximin MAP by developing tighter approximations as well as pursuing alternative search strategies such as best-first search or hybrids of depth-first and best-first search.

ACKNOWLEDGMENTS

F. G. Cozman is partially supported by CNPq grant Pq 305753/2022-3, and thanks the Center for Artificial Intelligence (C4AI-USP) with support from the São Paulo Research Foundation (FAPESP) grant 2019/07665-4) and from the IBM Corporation.

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

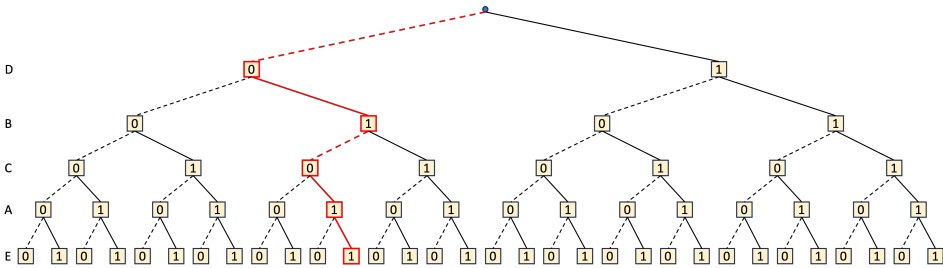

Figure 2: The OR search tree corresponding to the credal network from Figure 1a.

# A  APPENDIX

# B  DEPTH-FIRST SEARCH FOR MAXIMAX AND MAXIMIN MAP

The simplest approach to solve exactly the *maximax* and *maximin* MAP tasks in credal networks is to perform a depth-first search in the space of partial assignments to the variables (called the OR search space), and, for each complete assignment denoted by $\hat{\mathbf{x}}$, evaluate its score as the exact upper (resp., lower) probability $\overline{P}(\hat{\mathbf{x}}) = \prod_{i=1}^{n} \overline{P}(x_i|\pi_i)$ (resp. $\underline{P}(\hat{\mathbf{x}}) = \prod_{i=1}^{n} \underline{P}(x_i|\pi_i)$) where $x_i$ and $\pi_i$ are the values of $X_i$ and its parents $\Pi_i$ in $\hat{\mathbf{x}}$, respectively, and $\overline{P}(x_i|\pi_i) = \max \text{ext}(K(x_i|\pi_i))$ (resp. $\underline{P}(x_i|\pi_i) = \min \text{ext}(K(x_i|\pi_i))$). In this way, the optimal solution $\mathbf{x}^*$ corresponds to the assignment with the highest score (i.e., the maximum upper probability for maximax MAP, and maximum lower probability for maximin MAP, respectively). Although complete, the algorithm is inefficient because it enumerates all possible configurations of the variables. Therefore, its time complexity is bounded by $O(k^n)$, where $n$ is the number of variables and $k$ bounds their domain sizes, but it can operate in linear space (Pearl, 1984).

**Example 4.** *Figure 2 shows the OR search tree explored by the depth-first search algorithm when solving the maximax MAP task for the credal network in Figure 1a. A solution path corresponding to the assignment $\hat{\mathbf{x}} : (A = 1, B = 1, C = 0, D = 0, E = 1)$ is highlighted in red and its maximum MAP value is $g(\hat{\mathbf{x}}) = \overline{P}(A = 1) \cdot \overline{P}(B = 1|A = 1) \cdot \overline{P}(C = 0|A = 1) \cdot \overline{P}(D = 0|B = 1, C = 1) \cdot \overline{P}(E = 1|D = 0) = 0.21504$, where, for example, $\overline{P}(A = 1) = \max \text{ext}(K(A = 1)) = \max(\{0.6, 0.8\}) = 0.8$ and $\overline{P}(B = 1|A = 1) = \max \text{ext}(K(B = 1|A = 1)) = \max(\{0.5, 0.7, 0.5, 0.7\}) = 0.7$, respectively. The optimal maximax and maximin MAP solutions are in this case $\overline{\mathbf{x}}^* : (A = 0, B = 0, C = 1, D = 1, E = 0)$ with value $0.26244$ and $\underline{\mathbf{x}}^* : (A = 1, B = 1, C = 1, D = 1, E = 0)$ with value $0.0504$, respectively.*

# C  BUCKET ELIMINATION FOR MAXIMAX MAP

The Maximax MAP task defined by Equation 3 can be solved exactly using a bucket elimination procedure (Dechter, 1999) that extends the Credal Variable Elimination (CVE) algorithm developed for marginal inference in credal networks (Mauá & Cozman, 2020). The algorithm relies on the notion of a *potential* as well as combination and marginalization operators over potentials which are defined as follows.

**Definition 6** (potential). *Given a set of variables $\mathbf{Y}$, a* potential *$\phi(\mathbf{Y})$ is a set of non-negative real-valued functions $p(\mathbf{Y})$ on $\mathbf{Y}$. The product of two potentials $\phi(\mathbf{Y})$ and $\psi(\mathbf{Z})$ is defined by $\phi(\mathbf{Y}) \cdot \psi(\mathbf{Z}) = \{p \cdot q : p \in \phi(\mathbf{Y}), q \in \psi(\mathbf{Z})\}$. The max-marginal $\max_{\mathbf{Z}} \phi(\mathbf{Y})$ of a potential $\phi(\mathbf{Y})$ with respect to a subset of variables $\mathbf{Z} \subseteq \mathbf{Y}$ is defined by $\max_{\mathbf{Z}} \phi(\mathbf{Y}) = \{\max_{\mathbf{Z}} p(\mathbf{Y}) : p \in \phi(\mathbf{Y})\}$.*

Since the multiplication operator may grow the size of potentials dramatically, we introduce an additional pruning operation that can reduces the cardinality of a potential. Specifically, the operator $\max \phi(\mathbf{Y})$ returns the set of non-zero maximal elements of $\phi(\mathbf{Y})$, under the partial order $\leq$ defined component-wise as $p(\mathbf{Y}) \leq q(\mathbf{Y})$ iff $\forall \mathbf{y} \in \Omega_{\mathbf{Y}}, p(\mathbf{y}) \leq q(\mathbf{y})$, where $\Omega_{\mathbf{Y}}$ is the cartesian product of the domains of the variables in $\mathbf{Y}$: $\max \phi(\mathbf{Y}) = \{p(\mathbf{Y}) \in \phi(\mathbf{Y}) : \nexists q(\mathbf{Y}) \in \phi(\mathbf{Y}), p(\mathbf{Y}) \leq q(\mathbf{Y})\}$.

$$\phi(A) : \{p_1(A), p_2(A)\} \qquad\qquad \phi(A, B) : \{p_1(B|A), p_2(B|A), p_3(B|A), p_4(B|A)\}$$

| A | $p_1(A)$ |
|---|---|
| 0 | 0.4 |
| 1 | 0.6 |

| A | $p_2(A)$ |
|---|---|
| 0 | 0.2 |
| 1 | 0.8 |

| A | B | $p_1(B|A)$ |
|---|---|---|
| 0 | 0 | 0.9 |
| 0 | 1 | 0.1 |
| 1 | 0 | 0.5 |
| 1 | 1 | 0.5 |

| A | B | $p_2(B|A)$ |
|---|---|---|
| 0 | 0 | 0.9 |
| 0 | 1 | 0.1 |
| 1 | 0 | 0.3 |
| 1 | 1 | 0.7 |

| A | B | $p_3(B|A)$ |
|---|---|---|
| 0 | 0 | 0.7 |
| 0 | 1 | 0.3 |
| 1 | 0 | 0.5 |
| 1 | 1 | 0.5 |

| A | B | $p_4(B|A)$ |
|---|---|---|
| 0 | 0 | 0.7 |
| 0 | 1 | 0.3 |
| 1 | 0 | 0.3 |
| 1 | 1 | 0.7 |

Figure 3: Examples of potentials for the credal network from Figure 1a.

**Definition 7** (dominance). *Let $\phi(\mathbf{Y})$ and $\psi(\mathbf{Y})$ be two potentials defined on the subset of variables $\mathbf{Y}$. Then we say that $\phi(\mathbf{Y}) \leq \psi(\mathbf{Y})$ if and only if $\forall p(\mathbf{Y}) \in \phi(\mathbf{Y})$, $\exists q \in \psi(\mathbf{Y})$ such that $p(\mathbf{Y}) \leq q(\mathbf{Y})$, where the latter corresponds to component-wise $\leq$ defined above.*

**Proposition 1** (commuting $\max$ operators). *Let $\phi(X_i, \mathbf{X}_j)$ and $\psi(X_i, \mathbf{X}_k)$ be two potentials such that $\phi = \{p_1, p_2, \ldots, p_n\}$ and $psi = \{q_1, q_2, \ldots, q_m\}$. Then, the $\max$-marginal operator and the $\max$-pruning operator commute, and the following equality holds:*

$$\max_{X_i} \max_{P(\mathbf{Z}) \in K(\mathbf{Z})} \phi(X_i, \mathbf{X}_j) \cdot \psi(X_i, \mathbf{X}_k) = \max_{P(\mathbf{Z} \setminus \{X_i\}) \in K'(\mathbf{Z} \setminus \{X_i\})} \max_{X_i} \phi(X_i, \mathbf{X}_j) \cdot \psi(X_i, \mathbf{X}_k), \quad (5)$$

*where $\mathbf{Z} = \{X_i\} \cup \mathbf{X}_j \cup \mathbf{X}_k$, $K(\mathbf{Z})$ is the credal set for $\phi(X_i, \mathbf{X}_j) \cdot \psi(X_i, \mathbf{X}_k)$, and $K'(\mathbf{Z} \setminus \{X_i\})$ is the credal set for $\max_{X_i} \phi(X_i, \mathbf{X}_j) \cdot \psi(X_i, \mathbf{X}_k)$.*

*Proof.* Since $\max_{P(\mathbf{Z}) \in K(\mathbf{Z})}$ prunes the credal set by finding the dominating function,

$$\max_{P(\mathbf{Z}) \in K(\mathbf{Z})} \phi(X_i, \mathbf{X}_j) \cdot \psi(X_i, \mathbf{X}_k)$$

$$= \max_{P(\mathbf{Z} \setminus \mathbf{X}_k) \in K_\phi(\mathbf{Z} \setminus \mathbf{X}_k)} \phi(X_i, \mathbf{X}_j) \cdot \max_{P(\mathbf{Z} \setminus \mathbf{X}_j) \in K_\psi(\mathbf{Z} \setminus \mathbf{X}_j)} \psi(X_i, \mathbf{X}_k)$$

$$= p^*(X_i, \mathbf{X}_j) \cdot q^*(X_i, \mathbf{X}_k),$$

where $p^*(X_i, \mathbf{X}_j)$ and $q^*(X_i, \mathbf{X}_k)$ are the dominating function in the credal set $K_\phi(\mathbf{Z} \setminus \mathbf{X}_k)$ and $K_\psi(\mathbf{Z} \setminus \mathbf{X}_j)$, respectively.

By commuting $\max$-marginal operator,

$$\max_{X_i} \max_{P(\mathbf{Z}) \in K(\mathbf{Z})} \phi(X_i, \mathbf{X}_j) \psi(X_i, \mathbf{X}_k)$$

$$= \max_{P(\mathbf{Z} \setminus \{X_i\}) \in K'(\mathbf{Z} \setminus \{X_i\})} \max_{X_i} \phi(X_i, \mathbf{X}_j) \psi(X_i, \mathbf{X}_k)$$

$$= \max_{P(\mathbf{Z} \setminus \{X_i\}) \in K'(\mathbf{Z} \setminus \{X_i\})} \{\max_{X_i} p(X_i, \mathbf{X}_j) \cdot q(X_i, \mathbf{X}_k) | \forall p \in \phi, q \in \psi\}$$

$$= \max_{X_i} p^*(X_i, \mathbf{X}_j) \cdot q^*(X_i, \mathbf{X}_k).$$

$\square$

**Example 5.** *Consider again the credal network from Figure 1a. In Figure 3 we show the potentials $\phi(A)$ and $\phi(A, B)$ corresponding to the sets of extreme points of the local conditional credal sets $K(A)$ and $K(B|A)$, respectively. We can see that, for example, $\phi(A, B)$ has 4 extreme points represented by the distributions $p_1(B|A)$, $p_2(B|A)$, $p_3(B|A)$ and $p_4(B|A)$, respectively.*

Algorithm 3 describes the bucket elimination procedure called CBE that can be used to solve Equation 3. Let $o : X_1, X_2, \ldots, X_n$ be an ordering of the variables $\mathbf{X}$ such that $X_1$ is eliminated first, then $X_2$ and so on. First, the algorithm creates a set of potentials $\Gamma$ from the input local conditional credal sets $K(X_i|\Pi_i = \pi_{ij})$. Each potential $\phi_k$ contains the set of all conditional probability distributions $P(X_k|\Pi_k)$ such that $P(x_k|\pi_{kj}) = P(X_k = x_k|\Pi_k = \pi_{kj}) \in ext(K(X_k|\Pi_k = \pi_{kj}))$, where $\pi_{kj}$ is the $j$-th configuration of the variables $\Pi_k$.

The algorithm then eliminates each variable $X_k$ by maximization from the combination of potentials that contain $X_k$ in their scope, namely it computes a new potential $\lambda^k = \max \left( \max_{X_k} \prod_{\phi \in \Gamma_k} \phi \right)$. The resulting potential $\lambda^k$ is pruned by removing its non-maximal elements. Finally, the optimal maximax MAP value is obtained after eliminating the last variable in the ordering.

---

**Algorithm 3** Bucket Elimination for Maximax MAP

---

1: **procedure** CBE($\mathcal{C} = \langle \mathbf{X}, \mathbf{D}, \mathbf{K} \rangle$)
2:              ▷ Create the potentials
3:    Initialize $\Gamma \leftarrow \emptyset$
4:    **for all** variable $X_k \in \mathbf{X}$ **do**
5:      Let $\phi_k = \{p : p \in \text{ext}(K(X_k | \Pi_k))\}$
6:      Update $\Gamma = \Gamma \cup \{\phi_k\}$
7:    Create elimination ordering $o : X_1, \ldots, X_n$
8:    **for all** variable $X_k \in o$ **do**
9:             ▷ Create bucket $\Gamma_k$ for variable $X_k$
10:    Let $\Gamma_k = \{\phi : \phi \in \Gamma, X_k \in vars(\phi)\}$
11:    Update $\Gamma = \Gamma \setminus \Gamma_k$
12:         ▷ Compute the downward message
13:    Let $\lambda^k \leftarrow \max \left( \max_{X_k} \prod_{\phi \in \Gamma_k} \phi \right)$
14:    Update $\Gamma = \Gamma \cup \{\lambda^k\}$
15:    **return** $\max \left( \prod_{\phi \in \Gamma} \phi \right)$

---

$$\max_D \left( \max_E \max \left( \max_E \phi(D,E) \right) \cdot \max \left( \max_B \max \left( \max_C \phi(B,C,D) \cdot \max(\max_A(\phi(A) \cdot \phi(A,B) \cdot \phi(A,C))) \right) \right) \right)$$

Figure 4: Schematic bucket elimination for maximax MAP on the credal network from Figure 1a.

**Example 6.** *Figure 4 shows the schematic bucket elimination for maximax MAP on the credal network from Figure 1a. In this case, the variable ordering is: $o : A, C, B, E, D$. The intermediate potentials denoted by $\lambda$ are shown in red.*

**Theorem 3** (complexity). *Given a credal network $\mathcal{C} = \langle \mathbf{X}, \mathbf{D}, \mathbf{K} \rangle$, the CBE algorithm computes the optimal maximum MAP value of $\mathcal{C}$. The time and space complexity is bounded by $O(n \cdot M^2 \cdot d^{w^*})$, where $n$ is the number of variables, $d$ is the maximum domain size, and $M$ bounds the cardinality of the potentials.*

*Proof.* Clearly, the pruning operator $\max$ commutes with the max-marginalization operator in Equation 3. Therefore, eliminating first a variable and subsequently pruning the non-maximal elements from the resulting potential is equivalent to eliminating the variable from the maximizing distribution in Equation 3. □

### C.1 MINI-BUCKETS FOR MAXIMAX MAP

The CBE algorithm is exact for Maximax MAP but time and space exponential in the induced width of the credal network. We describe next a mini-bucket approximation for maximax MAP which we enhance further with a cost-shifting scheme based on moment matching.

Algorithm 4 and adapts the mini-bucket partitioning scheme developed for graphical models (Dechter & Rish, 2003) to the maximax MAP task in credal networks. Specifically, algorithm MB($i$) which approximates CBE is parameterized by an $i$-bound $i$ and works by partitioning large buckets into smaller subsets, called *mini-buckets*, each containing at most $i$ distinct variables. The mini-buckets are processed separately by maximizing out the bucket variable from the combination of potentials in the respective mini-bucket. Based on previous work (Dechter & Rish, 2003), it is possible to show that MB($i$) outputs an upper bound on the optimal maximax MAP value from Equation 3.

**Proposition 2** (mini-bucket bound). *Let $\phi(X_i, \mathbf{X}_j)$ and $\psi(X_i, \mathbf{X}_k)$ be two potentials such that $\phi = \{p_1, p_2, ..., p_n\}$ and $\psi = \{q_1, q_2, ..., q_m\}$, respectively. Then, the following inequality holds:*

---

**Algorithm 4** Mini-Buckets for Maximax MAP

---

1: **procedure** MB($\mathcal{C} = \langle \mathbf{X}, \mathbf{D}, \mathbf{K} \rangle$, $i$-bound)
2:    Initialize $\Gamma \leftarrow \emptyset$
3:    **for all** variable $X_k \in \mathbf{X}$ **do**
4:      Let $\phi_k = \{p : p \in ext(K(X_k | \Pi_k))\}$
5:      Update $\Gamma = \Gamma \cup \{\phi_k\}$
6:    Create elimination ordering $o : X_1, \ldots, X_n$
7:    **for all** variable $X_k \in o$ **do**
8:      ▷ Create bucket $\Gamma_k$ and mini-buckets $Q_{kr}$
9:      Let $\Gamma_k = \{\phi : \phi \in \Gamma, X_k \in vars(\phi)\}$
10:      Update $\Gamma = \Gamma \setminus \Gamma_k$
11:        ▷ Create mini-buckets $Q_{k1}, \ldots, Q_{kR}$
12:      Partition $\Gamma_k$ into $\{Q_{k1}, \ldots, Q_{kR}\}$
13:      **for all** $r = 1$ to $R$ **do**
14:        Let $\phi_{kr} = \prod_{\phi \in Q_{kr}} \phi$
15:        ▷ Compute the downward messages
16:      **for all** $r = 1$ to $R$ **do**
17:        Let $\lambda_r^k \leftarrow \max(\max_{X_k} \phi_{kr})$
18:        Update $\Gamma = \Gamma \cup \{\lambda_r^k\}$
19:    **return** $\max(\prod_{\phi \in \Gamma} \phi)$

---

A: $\phi(A)$   $\phi(A,B)$   $\phi(A,C)$

C:    $\phi(B,C,D)$   $\lambda^A(C)$

B: $\lambda^A(B)$    $\lambda^C(B,D)$

E:      $\phi(D,E)$

D: $\lambda^B(D)$    $\lambda^E(D)$

$\lambda^A(B) = \max(\max_A(\phi(A) \cdot \phi(A,B)))$

$\lambda^A(C) = \max(\max_A \phi(A,C))$

$\lambda^C(B,D) = \max(\max_C(\phi(B,C,D) \cdot \lambda^A(C)))$

$\lambda^B(D) = \max(\max_B(\lambda^A(B) \cdot \lambda^C(B,D)))$

$\lambda^E(D) = \max(\max_E(\phi(D,E)))$

Upper Bound $= \max(\max_D(\lambda^B(D) \cdot \lambda^E(D)))$

$$\max\left(\max_A\big(\phi(A) \cdot \phi(A,B) \cdot \phi(A,C)\big)\right) \le \max\left(\max_A(\phi(A) \cdot \phi(A,B)) \cdot \max_A \phi(A,C)\right)$$

Figure 5: Schematic execution of MB(2) on the credal network from Figure 1a

$$\max_{X_i}[\phi(X_i, \mathbf{X}_j) \cdot \psi(X_i, \mathbf{X}_k)] \le [\max_{X_i} \phi(X_i, \mathbf{X}_j)] \cdot [\max_{X_i} \psi(X_i, \mathbf{X}_k)] \tag{6}$$

*Proof.* Let $A = \max_{X_i} \phi(X_i, \mathbf{X}_j) \cdot \psi(X_i, \mathbf{X}_k)$ and let $a = \max_{X_i} p_t(X_i, \mathbf{X}_j) \cdot q_r(X_i, \mathbf{X}_j)$ be one of its components. Clearly, $\max_{X_i} p_t(X_i, \mathbf{X}_j) \cdot q_r(X_i, \mathbf{X}_j) \le \max_{X_i} p_t(X_i, \mathbf{X}_j) \cdot \max_{X_i} q_r(X_i, \mathbf{X}_j)$. Let $b = p_t^*(\mathbf{X}_j) = \max_{X_i} p_t(X_i, \mathbf{X}_j)$ and $c = q_r^*(\mathbf{X}_k) = \max_{X_i} q_r(X_i, \mathbf{X}_k)$ and let $B = \max_{X_i} \phi(X_i, \mathbf{X}_j)$ and $C = \max_{X_i} \psi(X_i, \mathbf{X}_k)$, respectively. Therefore, $a \le b \cdot c$. Then it follows that for every $a \in A$, we can identify an element of $a' \in B \cdot C$ such that $a \le a'$. □

**Example 7.** *Figure 5 shows the schematic execution of algorithm MB($i = 2$) on the credal network from Figure 1a. In this case, the elimination ordering is A, C, B, E, D, namely variable A is eliminated first, then C and so on. After eliminating the last variable D, we obtain an upper bound on the optimal maximax MAP value.*

**Proposition 3.** *Algorithm MB($i$) computes an upper bound on the optimal maximax MAP value.*

*Proof.* The result follows easily by applying Proposition 2. □

---

**Algorithm 5** Bucket Elimination for Maximin MAP

---

1: **procedure** CBE($\mathcal{C} = \langle \mathbf{X}, \mathbf{D}, \mathbf{K} \rangle$)
2:          ▷ Create the potentials
3:    Initialize $\Gamma \leftarrow \emptyset$
4:    **for all** variable $X_k \in \mathbf{X}$ **do**
5:      Let $\phi_k = \{p : p \in \text{ext}(K(X_k|\Pi_k))\}$
6:      Update $\Gamma = \Gamma \cup \{\phi_k\}$
7:    Create elimination ordering $o : X_1, \ldots, X_n$
8:    **for all** variable $X_k \in o$ **do**
9:           ▷ Create bucket $\Gamma_k$ for variable $X_k$
10:     Let $\Gamma_k = \{\phi : \phi \in \Gamma, X_k \in vars(\phi)\}$
11:     Update $\Gamma = \Gamma \setminus \Gamma_k$
12:        ▷ Compute the downward message
13:     Let $\lambda^k \leftarrow \min \left( \max_{X_k} \prod_{\phi \in \Gamma_k} \phi \right)$
14:     Update $\Gamma = \Gamma \cup \{\lambda^k\}$
15: **return** $\min \left( \prod_{\phi \in \Gamma} \phi \right)$

---

## D   BUCKET ELIMINATION FOR MAXIMIN MAP

For maximin MAP, we define a $\min$ pruning operator that returns the minimal elements of a potential $\phi(\mathbf{Y})$ relative to the same partial order $\leq$, namely $\min(\phi(\mathbf{Y})) = \{p(\mathbf{Y}) \in \phi(\mathbf{Y}) : \nexists q \in \phi(\mathbf{Y}), q(\mathbf{Y}) \leq p(\mathbf{Y})\}$.

**Proposition 4** (commuting $\max$-marginal and $\min$-pruning operators). *Let $\phi(X_i, \mathbf{X}_j)$ and $\psi(X_i, \mathbf{X}_k)$ be two potentials such that $\phi = \{p_1, p_2, \ldots, p_n\}$ and $psi = \{q_1, q_2, \ldots, q_m\}$. Then, commuting the $\max$-marginal operator and the $\min$-pruning operator yields the following inequality:*

$$\max_{X_i} \min_{P(\mathbf{Z}) \in K(\mathbf{Z})} \phi(X_i, \mathbf{X}_j) \cdot \psi(X_i, \mathbf{X}_k) \leq \min_{P(\mathbf{Z} \setminus \{X_i\}) \in K'(\mathbf{Z} \setminus \{X_i\})} \max_{X_i} \phi(X_i, \mathbf{X}_j) \cdot \psi(X_i, \mathbf{X}_k),$$

*where $\mathbf{Z} = \{X_i\} \cup \mathbf{X}_j \cup \mathbf{X}_k$, $K(\mathbf{Z})$ is the credal set for $\phi(X_i, \mathbf{X}_j) \cdot \psi(X_i, \mathbf{X}_k)$, and $K'(\mathbf{Z} \setminus \{X_i\})$ is the credal set for $\max_{X_i} \phi(X_i, \mathbf{X}_j) \cdot \psi(X_i, \mathbf{X}_k)$.*

*Proof.* The left-hand side of the inequalty can be written as,

$$\max_{X_i} \min_{P(\mathbf{Z}) \in K(\mathbf{Z})} \phi(X_i, \mathbf{X}_j) \cdot \psi(X_i, \mathbf{X}_k)$$
$$= \max_{X_i} \min_{P(\mathbf{Z} \setminus \mathbf{X}_k) \in K_\phi(\mathbf{Z} \setminus \mathbf{X}_k)} \phi(X_i, \mathbf{X}_j) \cdot \min_{P(\mathbf{Z} \setminus \mathbf{X}_j) \in K_\psi(\mathbf{Z} \setminus \mathbf{X}_j)} \psi(X_i, \mathbf{X}_k)$$
$$= \max_{X_i} p_*(X_i, \mathbf{X}_j) \cdot q_*(X_i, \mathbf{X}_k),$$

where $p_*(X_i, \mathbf{X}_j)$ and $q_*(X_i, \mathbf{X}_k)$ are the dominated function in the credal set $K_\phi(\mathbf{Z} \setminus \mathbf{X}_k)$ and $K_\psi(\mathbf{Z} \setminus \mathbf{X}_j)$, respectively.

By commuting $\max$-marginal and $\min$-pruning operator,

$$\min_{P(\mathbf{Z} \setminus \{X_i\}) \in K'(\mathbf{Z} \setminus \{X_i\})} \max_{X_i} \phi(X_i, \mathbf{X}_j) \psi(X_i, \mathbf{X}_k)$$
$$= \min_{P(\mathbf{Z} \setminus \{X_i\}) \in K'(\mathbf{Z} \setminus \{X_i\})} \{\max_{X_i} p(X_i, \mathbf{X}_j) \cdot q(X_i, \mathbf{X}_k) | \forall p \in \phi, q \in \psi\}$$
$$= r_*(\mathbf{X}_j, \mathbf{X}_k) \geq \max_{X_i} p_*(X_i, \mathbf{X}_j) \cdot q_*(X_i, \mathbf{X}_k),$$

where $r_*(\mathbf{X}_j, \mathbf{X}_k)$ is the dominated function in the set $\{\max_{X_i} p \cdot q | \forall p \in \phi, q \in \psi\}$. $\qquad\square$

Algorithm 5 describes the bucket elimination procedure called CBE that can be used to solve Equation 4. However, unlike maximax MAP, in this case CBE is no longer exact and only computes an upper bound on the optimal maximin MAP value. It is easy to see that $\max$ and $\min$ do not commute in Equation 4. We illustrate with a simple example that by pushing the outside $\max$ inside the $\min$ operator yields an upper bound: $\max(\min(3, 1), \min(3, 2)) = \max(1, 2) = 2 \leq \min(\max(3, 1), \max(3, 2)) = \min(3, 3) = 3$.

Let $o : X_1, X_2, \ldots, X_n$ be an ordering of the variables $\mathbf{X}$ such that $X_1$ is eliminated first, then $X_2$ and so on. First, the algorithm creates a set of potentials $\Gamma$ from the input local conditional credal sets $K(X_i|\Pi_i = \pi_{ik})$. Each potential $\phi_k$ contains the set of all conditional probability distributions $P(X_k|\Pi_k)$ such that $P(x_k|\pi_{kj}) = P(X_k = x_k|\Pi_k = \pi_{kj}) \in \text{ext}(K(X_k|\Pi_k = \pi_{kj}))$. The algorithm

Table 6: Results for Maximax MAP on 100 variables `random` networks. Average CPU time in seconds and number of nodes expanded using mini-bucket $i$-bounds from 2 to 12. Time limit 1 hour.

| size | algorithm | $i=2$ | | $i=4$ | | $i=6$ | | $i=8$ | | $i=10$ | | $i=12$ | |
|---|---|---|---|---|---|---|---|---|---|---|---|---|---|
| | | time | nodes | time | nodes | time | nodes | time | nodes | time | nodes | time | nodes |
| | | | | | | `random` credal networks | | | | | | | |
| | AOBB+MB($i$,1) | 2.90 | 29603 | 0.99 | 13934 | 0.54 | 6877 | 0.25 | 5280 | 0.18 | 2352 | 0.09 | 1438 |
| | AOBB+MB($i$,10) | 3.06 | 29603 | 1.47 | 13934 | 18.65 | 6877 | 501.36 | 5834 | 809.46 | 2582 | 1405.00 | 1931 |
| | AOBB+MB($i$,30) | 3.25 | 29603 | 1.72 | 13934 | 555.51 | 7609 | 2731.54 | 3462 | 3153.93 | 209 | - | - |
| | AOBB+MB($i$,50) | 3.25 | 29603 | 2.07 | 13934 | 558.29 | 7609 | 2781.15 | 2492 | 3284.57 | 251 | - | - |
| 100 | AOBB+MBMM($i$,1) | 2.25 | 21057 | 0.39 | 8316 | 0.40 | 4448 | 0.15 | 3544 | 0.20 | 1807 | 0.10 | 773 |
| | AOBB+MBMM($i$,10) | 1.97 | 21057 | 0.75 | 8316 | 15.99 | 4448 | 488.84 | 3906 | 797.66 | 1978 | 1770.41 | 1204 |
| | AOBB+MBMM($i$,30) | 2.12 | 21057 | 0.94 | 8316 | 541.91 | 4910 | 2708.84 | 2162 | 3189.16 | 105 | - | - |
| | AOBB+MBMM($i$,50) | 2.22 | 21057 | 1.04 | 8316 | 548.81 | 4910 | 2800.54 | 1633 | 3312.94 | 108 | - | - |
| | AOBB+MB($i$) | 2.87 | 29603 | 2.77 | 13934 | 972.37 | 7336 | - | - | - | - | - | - |

Table 7: Results for Maximax MAP on `random` credal networks. Average CPU time (sec) and number of nodes expanded using mini-bucket $i$-bounds from 2 to 12. Time limit 1 hour.

| size $(w^*,h)$ | algorithm | $i=2$ | | $i=4$ | | $i=6$ | | $i=8$ | | $i=10$ | | $i=12$ | |
|---|---|---|---|---|---|---|---|---|---|---|---|---|---|
| | | time | nodes | time | nodes | time | nodes | time | nodes | time | nodes | time | nodes |
| 20 (4,9) | DFS | 18.55 | 2097152 | | | | | | | | | | |
| | BB+MB($i$) | 0.01 | 204 | 0.01 | 49 | 774.78 | 22 | 775.56 | 22 | 776.53 | 22 | 774.82 | 22 |
| | BB+MB($i$,1) | 0.01 | 204 | 0.01 | 49 | 0.00 | 22 | 0.00 | 22 | 0.01 | 22 | 0.00 | 22 |
| | BB+MBMM($i$,1) | 0.01 | 58 | 0.01 | 24 | 0.00 | 22 | 0.00 | 22 | 0.01 | 22 | 0.00 | 22 |
| | AOBB+MB($i$) | 0.01 | 50 | 0.01 | 25 | 775.00 | 22 | 776.64 | 22 | 776.20 | 22 | 774.58 | 22 |
| | AOBB+MB($i$,1) | 0.01 | 50 | 0.01 | 25 | 0.01 | 22 | 0.01 | 22 | 0.01 | 22 | 0.01 | 22 |
| | AOBB+MBMM($i$,1) | 0.01 | 36 | 0.00 | 23 | 0.02 | 22 | 0.00 | 22 | 0.02 | 22 | 0.01 | 22 |
| 50 (9,17) | DFS | - | - | | | | | | | | | | |
| | BB+MB($i$) | 1.15 | 40301 | 0.57 | 9672 | 863.27 | 1009 | 3331.72 | 62 | - | - | - | - |
| | BB+MB($i$,1) | 1.08 | 40301 | 0.45 | 9672 | 0.05 | 947 | 0.01 | 117 | 0.02 | 57 | 0.02 | 52 |
| | BB+MBMM($i$,1) | 0.16 | 5178 | 0.07 | 818 | 0.02 | 276 | 0.01 | 56 | 0.02 | 52 | 0.02 | 52 |
| | AOBB+MB($i$) | 0.03 | 298 | 0.04 | 188 | 818.48 | 101 | 3333.34 | 55 | - | - | - | - |
| | AOBB+MB($i$,1) | 0.03 | 298 | 0.02 | 188 | 0.03 | 106 | 0.01 | 78 | 0.04 | 54 | 0.02 | 52 |
| | AOBB+MBMM($i$,1) | 0.03 | 169 | 0.01 | 107 | 0.06 | 94 | 0.01 | 53 | 0.07 | 52 | 0.02 | 52 |
| 100 (18,28) | DFS | - | - | | | | | | | | | | |
| | BB+MB($i$) | 3525.20 | 40136432 | 1014.36 | 6399801 | 1789.33 | 123229 | - | - | - | - | - | - |
| | BB+MB($i$,1) | 3444.32 | 46208835 | 576.28 | 6399801 | 21.88 | 348970 | 1.92 | 64725 | 0.39 | 10498 | 0.35 | 10254 |
| | BB+MBMM($i$,1) | 2481.97 | 34115031 | 154.77 | 1717539 | 29.81 | 362314 | 1.00 | 33382 | 0.86 | 15106 | 0.30 | 5205 |
| | AOBB+MB($i$) | 2.87 | 29603 | 2.77 | 13934 | 972.37 | 7336 | - | - | - | - | - | - |
| | AOBB+MB($i$,1) | 2.90 | 29603 | 0.99 | 13934 | 0.54 | 6877 | 0.25 | 5280 | 0.18 | 2352 | 0.09 | 1438 |
| | AOBB+MBMM($i$,1) | 2.25 | 21057 | 0.39 | 8316 | 0.40 | 4448 | 0.15 | 3544 | 0.20 | 1807 | 0.10 | 773 |
| 150 (27,38) | DFS | - | - | | | | | | | | | | |
| | BB+MB($i$) | - | - | | | | | | | | | | |
| | BB+MB($i$,1) | - | - | - | - | 2373.35 | 37994389 | 1408.44 | 21780102 | 1104.19 | 18280501 | 352.27 | 6721585 |
| | BB+MBMM($i$,1) | - | - | 3253.52 | 35114767 | 1119.96 | 15193688 | 1046.74 | 16149072 | 389.39 | 4938087 | 21.37 | 459902 |
| | AOBB+MB($i$) | 156.22 | 1452909 | 77.77 | 492383 | 1275.38 | 419109 | 3252.94 | 411225 | - | - | - | - |
| | AOBB+MB($i$,1) | 151.57 | 1452909 | 41.20 | 492383 | 30.97 | 351672 | 21.04 | 279360 | 22.17 | 281745 | 11.02 | 176409 |
| | AOBB+MBMM($i$,1) | 68.48 | 612721 | 23.95 | 323892 | 11.55 | 158230 | 6.56 | 148781 | 8.43 | 133786 | 2.62 | 47718 |
| 200 (36,48) | DFS | - | - | | | | | | | | | | |
| | BB+MB($i$) | - | - | - | - | - | - | - | - | - | - | - | - |
| | BB+MB($i$,1) | - | - | - | - | - | - | - | - | 3458.90 | 56460524 | 2639.46 | 58444711 |
| | BB+MBMM($i$,1) | - | - | - | - | - | - | 2780.10 | 41392520 | 2342.75 | 37828931 | 1495.49 | 29931863 |
| | AOBB+MB($i$) | 1555.37 | 8510209 | 1126.37 | 4969424 | 2285.98 | 813793 | 3458.96 | 96389 | - | - | - | - |
| | AOBB+MB($i$,1) | 1537.41 | 9957428 | 1112.31 | 7572102 | 979.05 | 9370560 | 738.42 | 7390775 | 433.13 | 3553045 | 113.94 | 1407589 |
| | AOBB+MBMM($i$,1) | 1155.54 | 8448489 | 1108.73 | 7985450 | 197.49 | 2193611 | 224.53 | 1974790 | 364.89 | 2768420 | 109.25 | 1344753 |

then eliminates each variable $X_k$ by maximization from the combination of potentials that contain $X_k$ in their scope, namely it computes a new potential $\lambda^k = \min \left( \max_{X_k} \prod_{\phi \in \Gamma_k} \phi \right)$. The resulting potential $\lambda^k$ is pruned by removing its non-minimal elements. After eliminating the last variable in the ordering, the resulting value is an upper bound on the optimal maximin MAP value.

# E  ADDITIONAL EXPERIMENTS

In this section we include additional experiments and details that were omitted from the main paper. We note that in all of our experiments, we did not consider any evidence.

## E.1  RESULTS FOR MAXIMAX MAP

Tables 7 and 8 summarize the results obtained on the `random`, and `grid` credal networks. The columns are indexed by the mini-bucket $i$-bound, and in each cell we show the average CPU time in seconds, and the average number of nodes expanded by the respective algorithm. We ran the OR and AND/OR branch and bound algorithms guided by mini-bucket heuristics without potential approximation and moment-matching (i.e., BB+MB(i), AOBB+MB(i)), mini-bucket heuristics with potential approximation of size 1 only (i.e., BB+MB(i,1), AOBB+MB(i,1)), and mini-bucket

Table 8: Results for Maximax MAP on `grid` credal networks. Average CPU time (sec) and number of nodes expanded using mini-bucket $i$-bounds from 2 to 12. Time limit 1 hour.

| size $(w^*,h)$ | algorithm | $i=2$ | | $i=4$ | | $i=6$ | | $i=8$ | | $i=10$ | | $i=12$ | |
|---|---|---|---|---|---|---|---|---|---|---|---|---|---|
| | | time | nodes | time | nodes | time | nodes | time | nodes | time | nodes | time | nodes |
| 25 (5,15) | DFS | 1051.72 | 67108864 | | | | | | | | | | |
| | BB+MB(i) | 0.06 | 3664 | 0.05 | 242 | 3275.18 | 27 | 3288.28 | 27 | 3283.06 | 27 | 3287.40 | 27 |
| | BB+MB(i,1) | 0.07 | 3664 | 0.01 | 242 | 0.00 | 27 | 0.01 | 27 | 0.01 | 27 | 0.01 | 27 |
| | BB+MBMM(i,1) | 0.06 | 1436 | 0.01 | 88 | 0.00 | 27 | 0.01 | 27 | 0.04 | 27 | 0.01 | 27 |
| | AOBB+MB(i) | 0.01 | 122 | 0.05 | 50 | 3278.58 | 27 | 3286.95 | 27 | 3283.82 | 27 | 3289.09 | 27 |
| | AOBB+MB(i,1) | 0.01 | 122 | 0.01 | 50 | 0.01 | 27 | 0.01 | 27 | 0.04 | 27 | 0.01 | 27 |
| | AOBB+MBMM(i,1) | 0.01 | 97 | 0.04 | 32 | 0.01 | 27 | 0.02 | 27 | 0.06 | 27 | 0.01 | 27 |
| 49 (9,25) | DFS | - | - | | | | | | | | | | |
| | BB+MB(i) | 141.09 | 4454273 | 5.10 | 53819 | 3524.41 | 498 | - | - | - | - | - | - |
| | BB+MB(i,1) | 189.40 | 4454273 | 2.23 | 53819 | 0.02 | 708 | 0.01 | 74 | 0.04 | 51 | 0.02 | 51 |
| | BB+MBMM(i,1) | 80.93 | 1728866 | 1.01 | 20768 | 0.01 | 96 | 0.02 | 53 | 0.06 | 51 | 0.03 | 51 |
| | AOBB+MB(i) | 0.03 | 562 | 0.11 | 266 | 3392.24 | 132 | - | - | - | - | - | - |
| | AOBB+MB(i,1) | 0.03 | 562 | 0.03 | 266 | 0.01 | 144 | 0.03 | 63 | 0.06 | 51 | 0.03 | 51 |
| | AOBB+MBMM(i,1) | 0.02 | 299 | 0.07 | 227 | 0.01 | 67 | 0.03 | 52 | 0.10 | 51 | 0.02 | 51 |
| 100 (14,38) | DFS | - | - | | | | | | | | | | |
| | BB+MB(i) | - | - | | | | | | | | | | |
| | BB+MB(i,1) | - | - | - | - | 2069.82 | 32793488 | 51.48 | 1342799 | 0.17 | 4123 | 0.06 | 783 |
| | BB+MBMM(i,1) | - | - | 3287.93 | 45186026 | 362.59 | 5259684 | 0.07 | 699 | 0.13 | 123 | 0.07 | 102 |
| | AOBB+MB(i) | 3.82 | 65648 | 0.23 | 2138 | - | - | - | - | - | - | - | - |
| | AOBB+MB(i,1) | 3.56 | 65648 | 0.13 | 2138 | 0.07 | 1930 | 0.09 | 1277 | 0.24 | 477 | 0.05 | 233 |
| | AOBB+MBMM(i,1) | 0.74 | 19531 | 0.30 | 2138 | 0.06 | 1235 | 0.07 | 226 | 0.21 | 107 | 0.05 | 102 |
| 144 (18,49) | DFS | - | - | | | | | | | | | | |
| | BB+MB(i) | - | - | - | - | - | - | - | - | - | - | - | - |
| | BB+MB(i,1) | - | - | - | - | - | - | 2882.01 | 51081184 | 986.25 | 23294052 | 5.44 | 184268 |
| | BB+MBMM(i,1) | - | - | - | - | 3244.43 | 38700863 | 38.68 | 736998 | 0.35 | 1810 | 0.16 | 1787 |
| | AOBB+MB(i) | 33.80 | 524460 | 0.42 | 8355 | - | - | - | - | - | - | - | - |
| | AOBB+MB(i,1) | 43.48 | 524460 | 0.46 | 8355 | 0.28 | 8324 | 0.33 | 7073 | 0.74 | 5710 | 0.15 | 2278 |
| | AOBB+MBMM(i,1) | 18.45 | 247311 | 0.73 | 8325 | 0.27 | 7413 | 0.11 | 1046 | 0.32 | 327 | 0.09 | 243 |
| 196 (20,57) | DFS | - | - | | | | | | | | | | |
| | BB+MB(i) | - | - | - | - | - | - | - | - | - | - | - | - |
| | BB+MB(i,1) | - | - | - | - | - | - | - | - | - | - | 2243.14 | 53048783 |
| | BB+MBMM(i,1) | - | - | - | - | - | - | 3240.10 | 39599609 | 1072.23 | 16685172 | 0.24 | 1060 |
| | AOBB+MB(i) | 961.58 | 10746465 | 1.54 | 32950 | 3478.82 | 32950 | - | - | - | - | - | - |
| | AOBB+MB(i,1) | 1162.59 | 10746465 | 1.92 | 32950 | 1.31 | 32950 | 1.62 | 32945 | 2.22 | 32950 | 1.01 | 25366 |
| | AOBB+MBMM(i,1) | 395.68 | 4201397 | 2.19 | 32950 | 1.03 | 31339 | 1.18 | 24508 | 1.83 | 32948 | 0.14 | 480 |

heuristics with potential approximation of size 1 and moment-matching (i.e., BB+MBMM(i,1), AOBB+MBMM(i,1)), respectively. In addition to the branch-and-bound algorithms, we also ran the brute force depth-first search algorithm denoted by DFS.

## E.2 RESULTS FOR MAXIMIN MAP

Tables 10 and 11 summarize the results obtained on the `random` and `grid` credal networks for the Maximin MAP task. In addition, Table 12 shows the results obtained on the real-world credal networks. As before, we report the average CPU time in seconds and average number of nodes expanded during search, across various mini-bucket $i$-bounds. We can see again that the AND/OR Branch and Bound algorithms that exploit the problem structure dramatically outperform their OR search counterparts, across all reported $i$-bounds.

However, unlike the Maximax MAP case, the Maximin MAP task appears to be much more difficult to solve by the proposed AND/OR search algorithm. This is primarily due to the much weaker mini-bucket heuristics compiled for Maximin MAP compared to those compiled for Maximax MAP. Indeed, we recall that the variable elimination procedure described by Algorithm 3 is not exact for Maximin MAP and only outputs an upper bound on the optimal Maximin MAP value. Consequently, the mini-bucket approximation of this bound turns out to be much looser even if we try to tighten it with the moment-matching scheme.

Consequently, the AND/OR branch and bound algorithms for Maximin MAP guided by the mini-bucket heuristics with potential approximation of size 1 and moment-matching can only solve random problems with up to 150 variables.

We observe however that AOBB+MB(i) with relatively small $i$-bounds (i.e., 2 or 4) performs quite well and is able to solve the problems relatively efficiently. This indicates that the corresponding mini-bucket bounds without potential approximation and moment-matching are tighter than those involving the Pareto least upper bound. Unfortunately, compiling the MB(i) heuristics for higher $i$-bounds is not feasible because of the computational overhead. Therefore, a possible direction of future work is to study of the mini-bucket heuristics for Maximin MAP and develop new ways to tighten them even further.

Table 9: Results for Maximax MAP on real-world credal networks. CPU time (sec) and number of nodes expanded using mini-bucket $i$-bounds from 2 to 12. Time limit 1 hour.

| instance (n, w, h) | algorithm | i = 2 | | i = 4 | | i = 6 | | i = 8 | | i = 10 | | i = 12 | |
|---|---|---|---|---|---|---|---|---|---|---|---|---|---|
| | | time | nodes | time | nodes | time | nodes | time | nodes | time | nodes | time | nodes |
| alarm (37,4,12) | BB+MB(i) | 2030.54 | 150170 | 3.98 | 50 | 5.67 | 39 | 6.07 | 39 | 3.87 | 39 | 9.33 | 39 |
| | BB+MBMM(i,1) | 5.52 | 4535 | 3.77 | 39 | 3.99 | 39 | 4.92 | 39 | 5.36 | 39 | 5.45 | 39 |
| | AOBB+MB(i) | 5.78 | 85 | 4.81 | 42 | 5.25 | 39 | 7.64 | 39 | 6.60 | 39 | 6.56 | 39 |
| | AOBB+MBMM(i,1) | 2.62 | 52 | 2.82 | 39 | 2.80 | 39 | 5.38 | 39 | 2.80 | 39 | 2.75 | 39 |
| child (20,3,6) | BB+MB(i) | 0.02 | 77 | 0.00 | 72 | 0.00 | 72 | 0.00 | 72 | 0.00 | 72 | 0.04 | 72 |
| | BB+MBMM(i,1) | 0.00 | 73 | 0.00 | 72 | 0.01 | 72 | 0.01 | 72 | 0.00 | 72 | 0.01 | 72 |
| | AOBB+MB(i) | 0.01 | 24 | 0.02 | 22 | 0.00 | 22 | 0.00 | 22 | 0.01 | 22 | 0.01 | 22 |
| | AOBB+MBMM(i,1) | 0.00 | 23 | 0.00 | 22 | 0.00 | 22 | 0.00 | 22 | 0.00 | 22 | 0.00 | 22 |
| hailfinder (56,5,11) | BB+MB(i) | - | - | 8.67 | 174 | - | - | - | - | - | - | - | - |
| | BB+MBMM(i,1) | - | - | 12.70 | 168 | 11.31 | 58 | 10.96 | 58 | 11.77 | 58 | 10.87 | 58 |
| | AOBB+MB(i) | 11.57 | 80 | 10.26 | 62 | - | - | - | - | - | - | - | - |
| | AOBB+MBMM(i,1) | 11.08 | 84 | 11.31 | 62 | 11.64 | 58 | 11.02 | 58 | 10.74 | 58 | 11.22 | 58 |
| insurance (27,7,11) | BB+MB(i) | 1.05 | 4829 | 1.12 | 4199 | - | - | - | - | - | - | - | - |
| | BB+MBMM(i,1) | 0.47 | 2230 | 0.13 | 534 | 0.12 | 250 | 0.09 | 170 | 0.07 | 170 | 0.10 | 170 |
| | AOBB+MB(i) | 0.08 | 89 | 0.08 | 86 | - | - | - | - | - | - | - | - |
| | AOBB+MBMM(i,1) | 0.07 | 87 | 0.07 | 78 | 0.06 | 57 | 0.06 | 56 | 0.07 | 56 | 0.08 | 56 |
| link (724,15,43) | BB+MB(i) | - | - | - | - | - | - | - | - | - | - | - | - |
| | BB+MBMM(i,1) | - | - | - | - | - | - | - | - | - | - | - | - |
| | AOBB+MB(i) | 23.09 | 67424 | 3.74 | 1772 | - | - | - | - | - | - | - | - |
| | AOBB+MBMM(i,1) | 9.77 | 33603 | 2.97 | 1004 | 2.99 | 978 | 2.79 | 978 | 2.58 | 793 | 2.88 | 735 |
| mastermind1 (1220,20,56) | BB+MB(i) | - | - | - | - | - | - | - | - | - | - | - | - |
| | BB+MBMM(i,1) | - | - | - | - | - | - | - | - | - | - | - | - |
| | AOBB+MB(i) | - | - | 102.60 | 34669 | - | - | - | - | - | - | - | - |
| | AOBB+MBMM(i,1) | - | - | 26.64 | 34493 | 9.96 | 17619 | 9.97 | 17619 | 9.83 | 17619 | 9.67 | 17619 |
| mastermind3 (3692,39,92) | BB+MB(i) | - | - | - | - | - | - | - | - | - | - | - | - |
| | BB+MBMM(i,1) | - | - | - | - | - | - | - | - | - | - | - | - |
| | AOBB+MB(i) | - | - | - | - | - | - | - | - | - | - | - | - |
| | AOBB+MBMM(i,1) | - | - | - | - | 3264.92 | 2180932 | 3088.55 | 2172466 | 3036.91 | 2167200 | 3010.41 | 2167117 |
| mildew (35,4,15) | BB+MB(i) | 58.77 | 349847 | 0.87 | 42 | 1.44 | 37 | 1.40 | 37 | 1.25 | 37 | 1.92 | 37 |
| | BB+MBMM(i,1) | 3.14 | 23006 | 0.10 | 37 | 0.11 | 37 | 0.11 | 37 | 0.11 | 37 | 0.12 | 37 |
| | AOBB+MB(i) | 0.13 | 112 | 0.73 | 41 | 1.85 | 37 | 1.81 | 37 | 1.27 | 37 | 1.44 | 37 |
| | AOBB+MBMM(i,1) | 0.08 | 67 | 0.13 | 37 | 0.11 | 37 | 0.11 | 37 | 0.12 | 37 | 0.12 | 37 |
| munin (1041,8,26) | BB+MB(i) | - | - | - | - | - | - | - | - | - | - | - | - |
| | BB+MBMM(i,1) | - | - | - | - | 35.85 | 88721 | 7.71 | 1056 | 7.47 | 1043 | 7.04 | 1043 |
| | AOBB+MB(i) | 2.11 | 1311 | 3.10 | 1076 | - | - | - | - | - | - | - | - |
| | AOBB+MBMM(i,1) | 1.96 | 1127 | 1.72 | 1076 | 1.78 | 1044 | 1.74 | 1043 | 1.62 | 1043 | 1.59 | 1043 |
| pedigree1 (334,21,47) | BB+MB(i) | - | - | - | - | - | - | - | - | - | - | - | - |
| | BB+MBMM(i,1) | - | - | - | - | - | - | - | - | - | - | 93.25 | 974000 |
| | AOBB+MB(i) | 1699.58 | 8389182 | 33.27 | 131390 | 158.79 | 131390 | - | - | - | - | - | - |
| | AOBB+MBMM(i,1) | 1543.91 | 8389051 | 34.84 | 131390 | 29.64 | 84817 | 29.90 | 104772 | 23.39 | 12259 | 22.50 | 466 |
| pedigree7 (1068,44,88) | BB+MB(i) | - | - | - | - | - | - | - | - | - | - | - | - |
| | BB+MBMM(i,1) | - | - | - | - | - | - | - | - | - | - | - | - |
| | AOBB+MB(i) | - | - | - | - | - | - | - | - | - | - | - | - |
| | AOBB+MBMM(i,1) | - | - | - | - | - | - | 2876.21 | 16663195 | 289.24 | 1907613 | 74.35 | 547430 |
| pedigree9 (1118,33,106) | BB+MB(i) | - | - | - | - | - | - | - | - | - | - | - | - |
| | BB+MBMM(i,1) | - | - | - | - | - | - | - | - | - | - | - | - |
| | AOBB+MB(i) | - | - | - | - | 3004.43 | 132421 | - | - | - | - | - | - |
| | AOBB+MBMM(i,1) | - | - | 3126.56 | 16782641 | 55.63 | 121021 | 12.69 | 25684 | 8.96 | 15477 | 16.43 | 32946 |
| xdiabetes (413,4,44) | BB+MB(i) | - | - | - | - | 165.72 | 415 | 119.47 | 415 | 191.26 | 415 | 166.74 | 415 |
| | BB+MBMM(i,1) | - | - | - | - | 0.52 | 415 | 0.56 | 415 | 0.58 | 415 | 0.57 | 415 |
| | AOBB+MB(i) | 0.40 | 529 | 1.04 | 426 | 158.02 | 415 | 179.66 | 415 | 163.06 | 415 | 114.08 | 415 |
| | AOBB+MBMM(i,1) | 0.19 | 495 | 0.12 | 426 | 0.21 | 415 | 0.21 | 415 | 0.26 | 415 | 0.25 | 415 |
| zbarley (48,7,21) | BB+MB(i) | 3255.20 | 589618 | 17.32 | 819 | - | - | - | - | - | - | - | - |
| | BB+MBMM(i,1) | 8.67 | 3897 | 7.89 | 74 | 15.71 | 50 | 15.82 | 50 | 16.92 | 50 | 17.35 | 50 |
| | AOBB+MB(i) | 22.24 | 170 | 13.44 | 76 | - | - | - | - | - | - | - | - |
| | AOBB+MBMM(i,1) | 7.97 | 82 | 7.66 | 57 | 15.53 | 50 | 16.11 | 50 | 16.99 | 50 | 16.93 | 50 |
| zpigs (441,10,25) | BB+MB(i) | - | - | 2578.16 | 3730772 | 223.00 | 1076 | - | - | - | - | - | - |
| | BB+MBMM(i,1) | - | - | 1.33 | 13235 | 0.79 | 463 | 0.69 | 443 | 0.73 | 443 | 0.77 | 443 |
| | AOBB+MB(i) | 0.37 | 535 | 0.64 | 463 | 232.95 | 456 | - | - | - | - | - | - |
| | AOBB+MBMM(i,1) | 0.19 | 480 | 0.16 | 454 | 0.21 | 444 | 0.35 | 443 | 0.33 | 443 | 0.35 | 443 |

## F   COMPARISON WITH LOCAL SEARCH ALGORITHMS

We also extended the local search algorithms developed previously for credal Marginal MAP (Marinescu et al., 2023) to solving the credal maximax and maximin MAP tasks as well. Specifically, we developed the following algorithms: Stochastic Local Search (SLS), Taboo Search (TS), Simulated Annealing (SA) and Guided Local Search (GLS), respectively. In all our experiments, we ran the algorithms for a total of 10 iterations (i.e., random restarts) with a maximum of 100,000 flips per iterations. The random flip probability was set to 0.1, the taboo list had a maximum size of 1,000, while the alpha and initial temperature used by SA were set to 0.1 and 100, respectively. As before, the time limit was set to 1 hour.

Tables 13 and 15 present the results for the Maximax MAP task on both random and real-world credal networks. Similarly, Tables 14 and 16 report the results for the Maximin MAP task on the same set of problem instances.

Table 10: Results for Maximin MAP on `random` credal networks. Average CPU time (sec) and number of nodes expanded using mini-bucket $i$-bounds from 2 to 12. Time limit 1 hour.

| size ($w^*$,$h$) | algorithm | $i=2$ | | $i=4$ | | $i=6$ | | $i=8$ | | $i=10$ | | $i=12$ | |
|---|---|---|---|---|---|---|---|---|---|---|---|---|---|
| | | time | nodes | time | nodes | time | nodes | time | nodes | time | nodes | time | nodes |
| 20 (4,9) | DFS | 18.55 | 2097152 | | | | | | | | | | |
| | BB+MB(i) | 0.10 | 2432 | 0.06 | 1453 | 0.05 | 1219 | 0.05 | 1219 | 0.05 | 1219 | 0.05 | 1219 |
| | BB+MB(i,1) | 0.10 | 2432 | 0.06 | 1453 | 0.05 | 1219 | 0.05 | 1219 | 0.05 | 1219 | 0.05 | 1219 |
| | BB+MBMM(i,1) | 0.10 | 1776 | 0.07 | 1268 | 0.11 | 1219 | 0.10 | 1219 | 0.20 | 1219 | 0.22 | 1219 |
| | AOBB+MB(i) | 0.01 | 62 | 0.01 | 48 | 769.29 | 41 | 783.35 | 40 | 781.64 | 41 | 781.84 | 41 |
| | AOBB+MB(i,1) | 0.01 | 93 | 0.01 | 73 | 0.00 | 69 | 0.00 | 69 | 0.00 | 69 | 0.00 | 69 |
| | AOBB+MBMM(i,1) | 0.01 | 85 | 0.01 | 71 | 0.01 | 69 | 0.01 | 69 | 0.01 | 69 | 0.01 | 69 |
| 50 (9,17) | DFS | - | - | - | - | - | - | - | - | - | - | - | - |
| | BB+MB(i) | 2197.76 | 50789183 | 2235.78 | 59543636 | 2466.74 | 49480309 | - | - | 2415.52 | 50355186 | 2383.77 | 48786553 |
| | BB+MB(i,1) | 2695.59 | 50966426 | 2374.55 | 47802338 | 2073.88 | 43649700 | 2307.14 | 48401873 | 2415.52 | 50355186 | 2383.77 | 48786553 |
| | BB+MBMM(i,1) | 2462.65 | 46982744 | 2166.87 | 43940482 | 2184.39 | 45607376 | 2405.55 | 50088103 | 2411.81 | 50116876 | 2390.75 | 48487153 |
| | AOBB+MB(i) | 0.03 | 511 | 0.05 | 407 | 1564.38 | 297 | 3257.63 | 591 | - | - | - | - |
| | AOBB+MB(i,1) | 0.06 | 992 | 0.04 | 767 | 0.03 | 674 | 0.04 | 697 | 0.04 | 777 | 0.03 | 683 |
| | AOBB+MBMM(i,1) | 0.06 | 816 | 0.05 | 678 | 0.04 | 610 | 0.06 | 702 | 0.06 | 689 | 0.05 | 683 |
| 100 (18,28) | DFS | - | - | - | - | - | - | - | - | - | - | - | - |
| | BB+MB(i) | - | - | - | - | - | - | - | - | - | - | - | - |
| | BB+MB(i,1) | - | - | - | - | - | - | - | - | - | - | - | - |
| | BB+MBMM(i,1) | - | - | - | - | - | - | - | - | - | - | - | - |
| | AOBB+MB(i) | 321.81 | 2288083 | 339.53 | 1490415 | 2335.24 | 686105 | - | - | - | - | - | - |
| | AOBB+MB(i,1) | 370.60 | 2519857 | 343.64 | 2403898 | 315.94 | 2380625 | 310.87 | 2385030 | 296.38 | 2348286 | 297.56 | 2356143 |
| | AOBB+MBMM(i,1) | 360.05 | 2510757 | 349.68 | 2488775 | 314.74 | 2381402 | 307.27 | 2380974 | 306.49 | 2370181 | 297.10 | 2363219 |
| 150 (27,38) | DFS | - | - | - | - | - | - | - | - | - | - | - | - |
| | BB+MB(i) | 3241.73 | 24284657 | 3244.87 | 25909874 | 3251.83 | 8188419 | - | - | 3240.74 | 30979810 | 3240.63 | 40077583 |
| | BB+MB(i,1) | 3241.71 | 23719408 | 3241.20 | 27381745 | 3241.03 | 30027153 | 3240.90 | 30044581 | 3240.69 | 33921616 | 3240.45 | 39299760 |
| | BB+MBMM(i,1) | 3241.64 | 23573902 | 3241.23 | 26682191 | 3241.07 | 29702609 | 3240.90 | 30399636 | 3240.69 | 33921616 | 3240.45 | 39299760 |
| | AOBB+MB(i) | 2536.15 | 10771808 | 2293.90 | 7957842 | 3248.25 | 1112588 | - | - | 2299.04 | 16106412 | 2450.80 | 20333769 |
| | AOBB+MB(i,1) | 3242.14 | 16470814 | 2498.93 | 15012036 | 2825.31 | 18955066 | 2990.98 | 19312663 | 2299.04 | 16106412 | 2450.80 | 20333769 |
| | AOBB+MBMM(i,1) | 3099.09 | 15395878 | 2951.84 | 16626578 | 2216.77 | 16053772 | 2299.99 | 15835166 | 2546.68 | 19344809 | 2605.07 | 19840368 |
| 200 (36,48) | DFS | - | - | - | - | - | - | - | - | - | - | - | - |
| | BB+MB(i) | - | - | - | - | - | - | - | - | - | - | - | - |
| | BB+MB(i,1) | - | - | - | - | - | - | - | - | - | - | - | - |
| | BB+MBMM(i,1) | - | - | - | - | - | - | - | - | - | - | - | - |
| | AOBB+MB(i) | - | - | - | - | - | - | - | - | - | - | - | - |
| | AOBB+MB(i,1) | - | - | - | - | - | - | - | - | - | - | - | - |
| | AOBB+MBMM(i,1) | - | - | - | - | - | - | - | - | - | - | - | - |

Table 11: Results for Maximin MAP on `grid` credal networks. Average CPU time (sec) and number of nodes expanded using mini-bucket $i$-bounds from 2 to 12. Time limit 1 hour.

| size ($w^*$,$h$) | algorithm | $i=2$ | | $i=4$ | | $i=6$ | | $i=8$ | | $i=10$ | | $i=12$ | |
|---|---|---|---|---|---|---|---|---|---|---|---|---|---|
| | | time | nodes | time | nodes | time | nodes | time | nodes | time | nodes | time | nodes |
| 25 (5,15) | DFS | 1051.72 | 67108864 | | | | | | | | | | |
| | BB+MB(i) | 0.06 | 3664 | 0.05 | 242 | 3275.18 | 27 | 3288.28 | 27 | 3283.06 | 27 | 3287.40 | 27 |
| | BB+MB(i,1) | 51.33 | 2139003 | 54.13 | 2101943 | 53.70 | 2083684 | 53.78 | 2083684 | 53.24 | 2083684 | 52.73 | 2083684 |
| | BB+MBMM(i,1) | 45.17 | 2142018 | 42.65 | 2092694 | 41.02 | 2083684 | 41.61 | 2083684 | 42.52 | 2083684 | 38.19 | 2083684 |
| | AOBB+MB(i) | 0.01 | 122 | 0.05 | 50 | 3278.58 | 27 | 3286.95 | 27 | 3283.82 | 27 | 3289.09 | 27 |
| | AOBB+MB(i,1) | 0.06 | 3680 | 0.06 | 3509 | 0.06 | 3496 | 0.05 | 3496 | 0.05 | 3496 | 0.05 | 3496 |
| | AOBB+MBMM(i,1) | 0.10 | 3653 | 0.10 | 3506 | 0.10 | 3496 | 0.10 | 3496 | 0.09 | 3496 | 0.09 | 3496 |
| 49 (9,25) | DFS | - | - | | | | | | | | | | |
| | BB+MB(i) | 141.09 | 4454273 | 5.10 | 53819 | 3524.41 | 498 | - | - | - | - | - | - |
| | BB+MB(i,1) | - | - | - | - | 3469.28 | 110049922 | 3555.44 | 129042447 | 3552.79 | 135249370 | 3579.86 | 141018196 |
| | BB+MBMM(i,1) | - | - | 3522.94 | 120903523 | 3593.77 | 130957532 | 2880.00 | 137553732 | - | - | - | - |
| | AOBB+MB(i) | 0.03 | 562 | 0.11 | 266 | 3392.24 | 132 | - | - | - | - | - | - |
| | AOBB+MB(i,1) | 1.30 | 65652 | 0.63 | 40579 | 0.92 | 39381 | 1.03 | 39748 | 1.02 | 39624 | 1.02 | 39624 |
| | AOBB+MBMM(i,1) | 2.76 | 67608 | 1.28 | 39705 | 1.26 | 40029 | 1.23 | 39650 | 1.20 | 39624 | 1.27 | 39624 |
| 100 (14,38) | DFS | - | - | | | | | | | | | | |
| | BB+MB(i) | 3410.80 | 45573423 | - | - | - | - | - | - | - | - | - | - |
| | BB+MB(i,1) | - | - | - | - | - | - | - | - | - | - | - | - |
| | BB+MBMM(i,1) | - | - | - | - | - | - | 2880.00 | 86787865 | - | - | - | - |
| | AOBB+MB(i) | 75.49 | 32883 | 79.54 | 1126 | 3428.01 | 102 | 3427.41 | 102 | - | - | - | - |
| | AOBB+MB(i,1) | 390.56 | 10949062 | 364.18 | 10424997 | 363.35 | 11333724 | 362.88 | 11541125 | 362.84 | 10914068 | 362.85 | 11132756 |
| | AOBB+MBMM(i,1) | 384.44 | 8520856 | 365.06 | 8545588 | 364.31 | 9051579 | 363.55 | 9057851 | 362.98 | 9007334 | 363.20 | 9076567 |
| 144 (20,57) | DFS | - | - | - | - | - | - | - | - | - | - | - | - |
| | BB+MB(i) | - | - | - | - | - | - | - | - | - | - | - | - |
| | BB+MB(i,1) | - | - | - | - | - | - | - | - | - | - | - | - |
| | BB+MBMM(i,1) | - | - | - | - | - | - | 2880.01 | 56234017 | - | - | - | - |
| | AOBB+MB(i) | 33.80 | 524460 | 0.42 | 8355 | - | - | - | - | - | - | - | - |
| | AOBB+MB(i,1) | - | - | 1897.99 | 61463739 | 1455.34 | 52054673 | 1455.18 | 52062183 | 1451.04 | 52765048 | 1451.10 | 55655798 |
| | AOBB+MBMM(i,1) | 3034.07 | 58071151 | 1729.59 | 45653311 | 1457.68 | 44054565 | 1455.40 | 44027607 | 1453.59 | 44223628 | 1452.31 | 48988157 |
| 196 (18,49) | DFS | - | - | - | - | - | - | - | - | - | - | - | - |
| | BB+MB(i) | - | - | - | - | - | - | - | - | - | - | - | - |
| | BB+MB(i,1) | - | - | - | - | - | - | - | - | - | - | - | - |
| | BB+MBMM(i,1) | - | - | - | - | - | - | 2880.02 | 58543756 | - | - | - | - |
| | AOBB+MB(i) | 961.58 | 10746465 | 1.54 | 32950 | 3478.82 | 32950 | - | - | - | - | - | - |
| | AOBB+MB(i,1) | - | - | 2279.84 | 31996332 | 2279.38 | 35298631 | 2248.19 | 38894634 | 1931.02 | 36055993 | - | - |
| | AOBB+MBMM(i,1) | - | - | 1670.52 | 20372342 | 1960.68 | 25768210 | 2259.95 | 32030876 | 1638.70 | 27250308 | 2249.21 | 37315455 |

# G  SUMMARY OF THE CONTRIBUTION

This paper presents significant advancements in the field of MAP inference for credal networks. While MAP inference has been extensively studied in Bayesian networks over the past decades, its counterpart in credal networks has received comparatively limited attention. To date, there exists no established algorithmic framework for solving credal MAP tasks in practical settings.

Table 12: Results for Maximin MAP on real-world credal networks. CPU time (sec) and number of nodes expanded using mini-bucket $i$-bounds from 2 to 12. Time limit 1 hour.

| instance $(n, w, h)$ | algorithm | $i=2$ time | nodes | $i=4$ time | nodes | $i=6$ time | nodes | $i=8$ time | nodes | $i=10$ time | nodes | $i=12$ time | nodes |
|---|---|---|---|---|---|---|---|---|---|---|---|---|---|
| alarm (37,4,12) | BB+MB($i$) | 128.78 | 2522160 | 110.17 | 2522160 | 96.03 | 2522160 | 98.37 | 2522160 | 77.19 | 2522160 | 81.93 | 2522160 |
|  | BB+MBMM($i$,1) | 126.35 | 2522160 | 106.60 | 2522160 | 101.46 | 2522160 | 97.39 | 2522160 | 89.28 | 2522160 | 63.16 | 2522160 |
|  | AOBB+MB($i$) | 6.90 | 460 | 3.86 | 348 | 6.79 | 348 | 5.60 | 348 | 5.68 | 348 | 5.49 | 348 |
|  | AOBB+MBMM($i$,1) | 6.96 | 394 | 3.77 | 348 | 7.11 | 348 | 6.64 | 348 | 6.17 | 348 | 6.91 | 348 |
| child (20,3,6) | BB+MB($i$) | 0.06 | 1792 | 0.04 | 1786 | 0.05 | 1786 | 0.08 | 1786 | 0.05 | 1786 | 0.04 | 1786 |
|  | BB+MBMM($i$,1) | 0.06 | 1790 | 0.05 | 1786 | 0.04 | 1786 | 0.04 | 1786 | 0.03 | 1786 | 0.04 | 1786 |
|  | AOBB+MB($i$) | 0.00 | 28 | 0.00 | 27 | 0.00 | 27 | 0.00 | 27 | 0.00 | 27 | 0.01 | 27 |
|  | AOBB+MBMM($i$,1) | 0.00 | 27 | 0.00 | 27 | 0.00 | 27 | 0.00 | 27 | 0.00 | 27 | 0.00 | 27 |
| hailfinder (56,5,11) | BB+MB($i$) | - | - | - | - | - | - | - | - | - | - | - | - |
|  | BB+MBMM($i$,1) | - | - | - | - | - | - | - | - | - | - | - | - |
|  | AOBB+MB($i$) | 10.79 | 411 | 10.24 | 389 | 10.61 | 389 | 8.66 | 389 | 9.63 | 389 | 9.64 | 389 |
|  | AOBB+MBMM($i$,1) | 10.47 | 408 | 10.14 | 389 | 10.29 | 389 | 10.07 | 389 | 10.09 | 389 | 9.37 | 389 |
| insurance (27,7,11) | BB+MB($i$) | 13.22 | 158023 | 14.32 | 155472 | 11.34 | 135472 | 11.14 | 135918 | 8.63 | 135918 | 12.57 | 135918 |
|  | BB+MBMM($i$,1) | 15.45 | 191919 | 13.12 | 140051 | 12.23 | 135918 | 10.79 | 135918 | 7.80 | 135918 | 12.63 | 135918 |
|  | AOBB+MB($i$) | 0.07 | 367 | 0.06 | 295 | 0.05 | 212 | 0.03 | 207 | 0.05 | 207 | 0.06 | 207 |
|  | AOBB+MBMM($i$,1) | 0.07 | 315 | 0.07 | 268 | 0.06 | 212 | 0.05 | 207 | 0.03 | 207 | 0.06 | 207 |
| link (724,15,43) | BB+MB($i$) | - | - | - | - | - | - | - | - | - | - | - | - |
|  | BB+MBMM($i$,1) | - | - | - | - | - | - | - | - | - | - | - | - |
|  | AOBB+MB($i$) | - | - | 1663.40 | 7820555 | 1245.14 | 7448824 | 1180.95 | 7427647 | 1139.88 | 7405547 | 1072.66 | 7405953 |
|  | AOBB+MBMM($i$,1) | - | - | 1538.40 | 7455392 | 1245.57 | 7406117 | 1188.28 | 7386176 | 1122.29 | 7383818 | 995.99 | 6862912 |
| mastermind1 (1220,20,56) | BB+MB($i$) | - | - | - | - | - | - | - | - | - | - | - | - |
|  | BB+MBMM($i$,1) | - | - | - | - | - | - | - | - | - | - | - | - |
|  | AOBB+MB($i$) | 49.00 | 64081 | 49.14 | 64240 | 34.42 | 62281 | 32.35 | 61822 | 28.90 | 61189 | 30.69 | 60236 |
|  | AOBB+MBMM($i$,1) | 47.79 | 64081 | 49.17 | 64259 | 34.04 | 62281 | 32.14 | 61737 | 32.29 | 61275 | 30.17 | 61245 |
| mastermind3 (3692,39,92) | BB+MB($i$) | - | - | - | - | - | - | - | - | - | - | - | - |
|  | BB+MBMM($i$,1) | - | - | - | - | - | - | - | - | - | - | - | - |
|  | AOBB+MB($i$) | - | - | - | - | 1200.00 | 1357913 | 1099.27 | 1357053 | 1092.61 | 1362794 | 1103.99 | 1370757 |
|  | AOBB+MBMM($i$,1) | - | - | - | - | 1205.83 | 1358178 | 1083.09 | 1360571 | 1077.06 | 1365064 | 1049.79 | 1367606 |
| mildew (35,4,15) | BB+MB($i$) | 0.09 | 5376 | 0.18 | 5376 | 0.17 | 5376 | 0.15 | 5376 | 0.10 | 5376 | 0.09 | 5376 |
|  | BB+MBMM($i$,1) | 0.15 | 5376 | 0.17 | 5376 | 0.19 | 5376 | 0.10 | 5376 | 0.19 | 5376 | 0.15 | 5376 |
|  | AOBB+MB($i$) | 0.22 | 1563 | 0.17 | 984 | 0.10 | 970 | 0.14 | 970 | 0.08 | 970 | 0.15 | 970 |
|  | AOBB+MBMM($i$,1) | 0.20 | 1128 | 0.19 | 972 | 0.15 | 970 | 0.21 | 970 | 0.20 | 970 | 0.12 | 970 |
| zpigs (441,10,25) | BB+MB($i$) | 0.58 | 2048 | 0.57 | 2048 | 0.51 | 2048 | 0.38 | 2048 | 0.32 | 2048 | 0.46 | 2048 |
|  | BB+MBMM($i$,1) | 0.56 | 2048 | 0.51 | 2048 | 0.48 | 2048 | 0.33 | 2048 | 0.47 | 2048 | 0.46 | 2048 |
|  | AOBB+MB($i$) | 0.58 | 2245 | 1.02 | 2192 | 0.88 | 2282 | 1.00 | 2282 | 0.88 | 2282 | 0.77 | 2282 |
|  | AOBB+MBMM($i$,1) | 0.62 | 2282 | 1.01 | 2282 | 0.75 | 2282 | 0.85 | 2282 | 0.90 | 2282 | 0.99 | 2282 |

Table 13: Results for Maximax MAP on `random` and `grid` networks. Average CPU time in seconds for systematic vs non-systematic search algorithms. Time limit 1 hour.

| size | AOBB+MBMM($i$,1) | SLS | TS | SA | GLS |
|---|---|---|---|---|---|
| `random` networks | | | | | |
| 20 | **0.00** | 49.60 | 46.34 | 33.61 | 55.83 |
| 50 | **0.01** | 184.46 | 107.5 | 98.24 | 175.69 |
| 100 | **0.10** | 372.75 | 188.92 | 196.96 | 352.78 |
| 150 | **2.62** | 565.95 | 223.03 | 300.46 | 529.20 |
| 200 | **109.25** | 681.60 | 438.32 | 326.83 | 563.61 |
| `grid` networks | | | | | |
| 25 | **0.01** | 53.21 | 44.77 | 38.20 | 59.95 |
| 49 | **0.01** | 186.56 | 68.27 | 59.30 | 169.44 |
| 100 | **0.05** | 350.69 | 171.27 | 164.97 | 327.31 |
| 144 | **0.09** | 421.20 | 203.42 | 207.58 | 424.90 |
| 196 | **0.14** | 572.15 | 312.78 | 362.54 | 456.23 |

Recently, Marinescu et al. (2023) pioneered the study of Marginal MAP inference in credal networks – a generalization of pure MAP inference. They introduced several stochastic local search algorithms alongside an exact brute-force depth-first search method. However, their empirical evaluation revealed that these approaches are either limited to very small problem instances or lack guarantees regarding the quality of the solutions produced.

In response to these limitations, we propose a novel branch-and-bound search framework designed to address two critical challenges: (1) scalability to larger and more complex credal networks, and (2) provision of solution quality guarantees, particularly optimality. Our approach leverages the AND/OR search space to exploit the underlying problem structure efficiently. This is further enhanced by a partitioning-based heuristic that integrates potential approximations with cost-shifting strategies. The

Table 14: Results for Maximin MAP on `random` and `grid` networks. Average CPU time in seconds for systematic vs non-systematic search algorithms. Time limit 1 hour.

| size | AOBB+MBMM($i$, 1) | SLS | TS | SA | GLS |
|------|------------------|-----|----|----|----|
| | random networks | | | | |
| 20 | **0.01** | 51.34 | 49.47 | 35.47 | 59.17 |
| 50 | **0.04** | 232.09 | 103.47 | 72.94 | 139.67 |
| 100 | 297.10 | 349.30 | 175.27 | **149.91** | 272.30 |
| 150 | 2216.77 | 471.92 | 311.40 | **198.21** | 384.21 |
| 200 | - | 576.57 | 380.96 | **262.67** | 435.96 |
| | grid networks | | | | |
| 25 | **0.09** | 61.80 | 46.19 | 39.46 | 62.67 |
| 49 | **1.20** | 178.29 | 73.36 | 63.61 | 120.19 |
| 100 | 362.98 | 355.03 | 213.00 | **114.30** | 258.98 |
| 144 | 1452.31 | 401.44 | 257.59 | **171.27** | 327.11 |
| 196 | 1638.70 | 508.97 | 277.42 | **245.27** | 466.84 |

Table 15: Results for Maximax MAP on the real-world credal networks. CPU time in seconds for systematic vs non-systematic search algorithms. Time limit 1 hour.

| instance | AOBB+MBMM($i$,1) | SLS | TS | SA | GLS |
|----------|------------------|-----|----|----|----|
| alarm | **2.62** | 3600.05 | 3600.00 | 3600.02 | 3600.02 |
| child | **0.00** | 23.22 | 28.96 | 8.88 | 14.99 |
| hailfinder | **10.74** | 3600.05 | 3600.05 | 3600.05 | 3600.05 |
| insurance | **0.06** | 296.68 | 219.52 | 146.25 | 145.71 |
| link | **2.58** | 3600.01 | 3600.01 | 3600.01 | 3600.01 |
| mastermind1 | **9.67** | 3543.37 | 874.84 | 1056.13 | 2629.90 |
| mastermind3 | **3010.41** | 3600.02 | 3600.02 | 3600.02 | 3600.02 |
| mildew | **0.08** | 242.49 | 128.90 | 124.13 | 257.74 |
| munin | **1.59** | 3600.00 | 1678.61 | 2563.09 | 3600.01 |
| pedigree1 | **22.50** | 3600.00 | 3600.25 | 3600.25 | 3600.25 |
| pedigree7 | **74.35** | 3600.00 | 3600.25 | 3600.25 | 3600.25 |
| pedigree9 | **8.96** | 3600.00 | 3600.25 | 3600.25 | 3600.25 |
| xdiabetes | **0.12** | 445.32 | 726.82 | 622.27 | 688.75 |
| zbarley | **7.66** | 3600.08 | 3600.08 | 3600.08 | 3600.08 |
| zpigs | **0.19** | 2478.80 | 638.01 | 673.14 | 1813.44 |

AND/OR search space, previously shown to yield substantial time savings in Bayesian networks, is here extended to credal networks and to both maximax and maximin MAP tasks.

Given that mini-bucket approximations of variable elimination in credal networks often incur high computational costs due to very large potentials, we introduce a novel approximation scheme. This scheme utilizes the Pareto least upper bound concept for multi-dimensional vectors to manage potential complexity effectively.

Our empirical results obtained on both synthetic and more realistic credal networks demonstrate that the proposed methods not only enhance computational efficiency but also scale to large networks with over 1,000 variables, all while guaranteeing the optimality of the solutions.

Finally, we observed that Maximin MAP is much more difficult to solve by our proposed algorithms than Maximax MAP. This is because the mini-bucket based heuristic upper bounds for Maximin MAP are significantly weaker than those compiled for Maximax MAP. Therefore, another avenue for future work is to explore new ways to tighten the mini-bucket heuristics for Maximim MAP.

Table 16: Results for Maximin MAP on the real-world credal networks. CPU time in seconds for systematic vs non-systematic search algorithms. Time limit 1 hour.

| instance | AOBB+MBMM($i$,1) | SLS | TS | SA | GLS |
|---|---|---|---|---|---|
| alarm | **3.77** | 3600.05 | 3600.00 | 3600.02 | 3600.02 |
| child | **0.00** | 25.54 | 19.07 | 6.43 | 31.67 |
| hailfinder | **9.37** | 3600.15 | 3600.04 | 3600.06 | 3600.05 |
| insurance | **0.03** | 447.66 | 113.53 | 98.23 | 196.98 |
| link | **995.99** | 3600.01 | 3600.02 | 3600.01 | 3600.03 |
| mastermind1 | **30.17** | 1965.99 | 948.96 | 1237.18 | 2295.69 |
| mastermind3 | **1049.79** | 3600.02 | 3600.02 | 3076.61 | 3600.02 |
| mildew | **0.12** | 741.58 | 557.08 | 190.81 | 240.82 |
| zpigs | **0.62** | 894.96 | 282.35 | 257.57 | 542.11 |

