# OpenReview forum: "Branch and Bound Search for Exact MAP Inference in Credal Networks"
_ICLR.cc/2026/Conference — ICLR 2026 Poster_

### Official Review · Reviewer_jvi6 · 2025-10-18

**Soundness:** 3
**Presentation:** 3
**Contribution:** 3
**Rating:** 8
**Confidence:** 4

**Summary:**

The authors propose a new way of exactly deriving versions of the MAP estimate for credal networks, a generalization of Bayesian networks to sets of probabilities. They provide experimental evidence testifying the effectiveness of their method.

**Strengths:**

The paper is well written, well motivated, it solves an open problem, and it does so exactly. The experiments show that the proposed method is indeed effective.

**Weaknesses:**

This is a strong paper, with only two minor -- almost cosmetic -- weaknesses, and it deserves to be accepted.

(i) There are a few typos, e.g. in Line 2, it should be "Instead of precise [...]"; there are two parentheses in the second line of section 2, so it is displayed $P_i = P(X_i | \Pi_i))$

(ii) The importance of causal inference in general, and credal networks in particular, for the field of Imprecise Probabilistic Machine Learning could be briefly mentioned in the related work section.

**Questions:**

See weaknesses.

---

> ### Author Response · Authors · 2025-11-18
> **Rebuttal**
>
> We sincerely thank the reviewer for their insightful and constructive feedback.
>
> * W1: We will fix the typos and improve the overall presentation.
>
> * W2: We will expand the related work section to include a discussion about application of credal networks to causal reasoning. Indeed, following previous work by [Zaffalon et al, 2020], our proposed search algorithms for credal MAP can be readily used to explain interventions in quasi-Markovian structural causal models.
>
> [Zaffalon et al, 2020] M. Zaffalon, A. Antonucci, R. Cabanas. Structural Causal Models Are (Solvable by) Credal Networks. In PGM 2020.

---

### Official Review · Reviewer_KRmC · 2025-10-18

**Soundness:** 3
**Presentation:** 3
**Contribution:** 3
**Rating:** 8
**Confidence:** 3

**Summary:**

This paper presents new depth-first branch-and-bound algorithms for performing exact Maximum a Posteriori (MAP) inference in credal networks—a generalization of Bayesian networks that allow imprecise probabilities.
The authors introduce two related inference tasks, maximax and maximin MAP, and develop algorithms that leverage an AND/OR search space to exploit problem decomposition.
They also propose a partitioning-based heuristic enhanced with a cost-shifting (moment-matching) strategy to guide the search.
Extensive experiments on both synthetic and real-world credal networks demonstrate significant efficiency improvements and scalability up to models with over 3000 variables.

**Strengths:**

The paper makes a meaningful step forward by providing the first exact branch-and-bound framework for MAP inference in credal networks, addressing a clear gap in the literature.

The proposed AND/OR Branch-and-Bound (AOBB) approach is well-motivated, theoretically grounded, and effectively extends existing frameworks from Bayesian networks to the more general credal setting.

The introduction of partitioning-based mini-bucket bounds with moment-matching (MBMM) is both technically interesting and empirically effective, improving heuristic accuracy without excessive computational cost.

**Weaknesses:**

While the paper acknowledges weaker heuristics and higher computational difficulty for the maximin case, the discussion could better analyze why this occurs and suggest concrete mitigation strategies.

The theoretical complexity results are concise but could benefit from a more intuitive discussion of practical bottlenecks—especially regarding heuristic pre-compilation and space trade-offs.

**Questions:**

How sensitive is the algorithm’s performance to the choice of pseudotree structure? Could dynamic variable ordering further enhance pruning efficiency?

For the maximin MAP task, have you explored any alternative bounding schemes beyond mini-buckets that could yield tighter lower bounds?

Could the proposed AOBB framework be adapted for anytime inference (as hinted in Section 5), and if so, how would heuristic accuracy affect the anytime performance curve?

---

> ### Author Response · Authors · 2025-11-18
> **Rebuttal**
>
> We sincerely thank the reviewer for their insightful and constructive feedback. Below, we address each of your comments and questions with detailed responses.
>
> * Q1: Yes, the quality of the pseudo tree does have an impact on the algorithm’s performance. For depth-first search, shallower pseudo-trees with many branches that capture problem decomposition better are preferred. Dynamic variable ordering can also help improve performance as previously observed in the case of Bayesian networks. However, the mini-bucket heuristic used to guide the search algorithms in this paper is pre-compiled along a given elimination order and this restricts the search to be conducted along a static variable ordering. We will clarify this issue in the paper.
>
> * Q2: We are currently working on extending the linear programming based approach employed by the ApproxLP algorithm [Antonucci et al, 2014] for marginal inference in credal networks to computing upper bounds for both maximax and maximin MAP.
>
> * Q3: As currently implemented, our AOBB is already an anytime search algorithm. However, it is not the most efficient one (that is because AOBB moves to exploring another branch of the pseudo tree only after it fully explores the search space corresponding to the current branch in the pseudo tree). To make it more efficient, we can incorporate, for example, the breadth-rotating mechanism developed by Otten and Dechter (2011). This mechanism enables the algorithm to perform simultaneous depth-first explorations across multiple branches of the pseudo tree. Furthermore, the accuracy of the heuristic which is typically associated with higher i-bound values (e.g., Tables 1, 2) significantly impacts the performance: the more accurate the heuristics, the faster the algorithm is likely to converge to the optimal solution. Therefore, we expect the anytime curves to converge faster with stronger mini-bucket heuristics. We will include additional results and an analysis of the anytime behavior.
>
> [Antonucci et al, 2014] A. Antonucci, C. de Campos, D. Huber, M. Zaffalon. Approximate credal network updating by linear programming with applications to decision making. IJAR 58 (2014), pages 25-38.

---

### Official Review · Reviewer_Q5Sg · 2025-10-30

**Soundness:** 4
**Presentation:** 3
**Contribution:** 2
**Rating:** 6
**Confidence:** 3

**Summary:**

This paper focuses on Maximum a Posteriori (MAP) inference in credal networks, which generalize Bayesian networks by allowing imprecise probabilities represented through convex sets of distributions (credal sets).

The authors define two MAP tasks, maximax MAP (finding assignments maximizing the upper probability) and maximin MAP (maximizing the lower probability). To solve these tasks, the paper introduces:

1- Depth-first Branch-and-Bound algorithms for exact MAP inference in credal networks, extending prior AND/OR search formulations used for Bayesian MAP inference.

2- A partitioning-based mini-bucket heuristic with potential approximation and cost-shifting (moment matching) to guide the search and reduce runtime.

The algorithms are evaluated on random and real-world credal networks (e.g., ALARM, Link, Mastermind) and demonstrate scalability to networks with over 3,000 variables.

**Strengths:**

1- The paper is mathematically sound, with clear definitions of credal networks, the maximax/maximin MAP formulations, and the bounding procedures.

2- Well-written and organized, following the style of classical graphical model research.

3- Empirical results demonstrate strong performance improvements over simpler search strategies and show good scalability.

4- Provides one of the few systematic attempts to perform exact inference in credal networks, which are otherwise rarely explored.

**Weaknesses:**

1- The overall contribution is incremental, mainly extending existing AND/OR branch-and-bound frameworks and mini-bucket heuristics from Bayesian to credal networks. The adaptation is conceptually straightforward, and the heuristic improvements are mostly engineering refinements rather than theoretical advances.

2- The runtime reduction relies on heuristic approximations (mini-buckets, Pareto least upper bounds, and moment matching). These are not guaranteed to always yield efficient pruning or optimal bounds, and no theoretical runtime guarantees are provided.

3- While the reported results are promising, they are mostly demonstrated on synthetic credal networks. Extending the experiments to more practical domains could further validate the effectiveness and real-world applicability of the proposed approach.

4- The claim that the proposed method “solves these tasks exactly in practice” is somewhat ambiguous. In principle, the algorithm can find the true optimal MAP solution if given enough time, since branch-and-bound guarantees exactness through exhaustive search with pruning. However, in practice, the approach relies heavily on heuristic bounds to reduce runtime, meaning that its “exactness” depends on the tightness of those bounds. This makes the statement somewhat overstated and potentially misleading without a clearer discussion of the trade-off between theoretical exactness and practical efficiency.

**Questions:**

See weaknesses.

---

> ### Author Response · Authors · 2025-11-18
> **Rebuttal**
>
> We sincerely thank the reviewer for their insightful and constructive feedback. Below, we address each of your comments and questions with detailed responses.
>
> * W1: We acknowledge that the proposed mini-bucket and AND/OR search algorithms build upon schemes originally developed for Bayesian networks. However, to the best of our knowledge, this work represents the first exploration and evaluation of such algorithms in the context of credal networks. We believe this constitutes a meaningful and at the same time valuable contribution to the AI community.
>
> * W2: Our mini-bucket heuristics guarantee upper bounds for both maximin and maximax partial MAP assignments. This translates into admissible heuristics that guarantee the optimality of the proposed depth-first search scheme because we are solving a maximization problem. Theorem 2 in Section 4 establishes the time and space complexity of the mini-bucket heuristic. Furthermore, the mini-bucket bounds get tighter (more accurate) as the mini-bucket i-bound increases – the size of the search space gets smaller at higher i-bounds (this is illustrated for example by the number of nodes shown in all our Tables from Section 5).
>
> * W3: Our empirical evaluation does include realistic credal networks. Specifically, the problem instances in Tables 3 and 4 are based on real-world Bayesian networks where we relaxed the original conditional probability values into probability intervals. In addition, we will consider adding more results with other realistic networks such as those developed by [Antonucci et al, 2009].
>
> * W4: In our experiments, the proposed search algorithms are able to prove optimality within the allotted time limit (1 hour). Therefore, we will clarify this issue in the experimental section.
>
> [Antonucci et al., 2009] A. Antonucci, R. Brühlmann, A. Piatti, M. Zaffalon Credal networks for military identification problems. In IJAR 50 (2009) pages 666-679.

---

### Official Review · Reviewer_JLKE · 2025-10-31

**Soundness:** 3
**Presentation:** 2
**Contribution:** 3
**Rating:** 6
**Confidence:** 3

**Summary:**

This paper uses various techniques developed that have been previously applied to MAP inference in Bayesian networks, and extends them to credal networks, specifically to the maximax and maximin MAP tasks. The resulting algorithm is provably exact and empirically more efficient than existing algorithms (both exact and not).

**Strengths:**

The algorithm is a substantial improvement over existing work, in a challenging and significant problem. The theoretical quality seems to be very high. The work is mostly clearly presented, though improvement is possible there.

**Weaknesses:**

The appendices include a lot of additional material, but the individual appendices are not referenced from the main text, making it an "exercise for the reader" to discover when they should skip from the main text to an appendix. The storyline of the main paper came across as incoherent in section 4 due to this.

**Questions:**

1. Section 1 mentioned Marginal MAP, but doesn't define it. How does it relate to the MAP tasks studied in this paper: Is one harder than the other? For which real-world problems are they best-suited?
2. Figure 1(b): according to the definition, it seems there should be an arc between C and D, correct? Does this affect the solution tree or the way the algorithms operate on this example?
3. Some questions about Algorithm 1:
  - there seems to be a hat missing on $\mathbf{x}_k$ in line 12 of the algorithm (similarly, line 216 of the text has a bar instead of a hat);
  - $S$ is not mentioned anywhere else in the algorithm. Is it an alias for $v(s)$? If so, I suggest to write that instead.
4. What is the distribution of the \texttt{random} graphs?
5. According to the main text, the algorithm for maximin MAP is much less efficient than the maximax one. Yet when comparing mastermind3 in tables 3 and 4, the maximin case is much faster. What is going on here?

### Minor comments

- line 27: "Instead *of*"
- definition of $\max$ on line 251: this should rule out the possibility of taking $q(Y)$ the same as $p(Y)$; as written, $\max$ is always the empty set. Same for $\min$ (line 310)
- the two-column form of the algorithms makes it impossible to see at which nesting level the first line of the second column is. Please resolve this, for instance by adding vertical helper lines.
- section "A Appendix" is empty; remove/replace this header
- Definition 6 and the text below it in Appendix C replicates part of the main text
- line 748: stray "and"

---

> ### Author Response · Authors · 2025-11-18
> **Rebuttal**
>
> We sincerely thank the reviewer for their insightful and constructive feedback. Below, we address each of your comments and questions with detailed responses.
>
> * Q1: The Marginal MAP tasks in credal networks were first defined in [Marinescu et al, 2023] and call for finding the assignment to a subset of variables that has the highest upper or lower marginal probability. Therefore, credal Marginal MAP generalizes credal MAP studied in this paper, however it is much more difficult to solve because the task involves both maximization and summation. Specifically, evaluating a Marginal MAP assignment involves summing out (marginalizing out) the remaining variables. Credal Marginal MAP is relevant to applications where there are hidden variables that cannot be observed such as the probabilistic diagnosis and intelligence reports analysis problems suggested in [Marinescu et al, 2023]. We will address the distinction between the two tasks more precisely in the paper.
>
> * Q2: That’s correct, there should be a back-arc between C and D. We will fix the figure. It does not affect how the algorithm works.
>
> * Q3: Yes, there is a missing hat there. The partial solution tree should have been denoted by $\hat{\textbf{x}}_k$ throughout the paper. We will fix this issue.
>
> * Q4: We assumed a uniform distribution. Specifically, our random networks were generated as follows: for each node we selected two other nodes uniformly at random to act as parents while ensuring that the graph is a DAG. The conditional probability distributions were also generated uniformly at random. We will expand the experimental section with more details about the benchmark networks we used.
>
> * Q5: The mastermind networks contain a significant amount of determinism i.e., zero-probability tuples. The search spaces explored by maximax and maximin algorithms are very different, and the discrepancy observed is likely caused by determinism, namely the zero-probability partial assignments lead to early pruning.
>
> Finally, we will carefully revise the main paper and appendix to fix all the issues raised by the reviewer in the minor comments section. We will also ensure that Section 4 refers to the appropriate sections of the appendix (for additional details and/or results).

---

### Meta-Review · Area_Chair_RefA · 2025-12-20

**Summary:**

The paper describes branch and bound algorithms for maximax and maximin MAP inference in credal networks.  The reviewers raised the following concerns:

1. The overall contribution is incremental
2. No guaranteed runtime due to the use of heuristics
3. The claim of exact solution is overstated

However those concerns were not deal breakers as the reviewers unanimously recommended acceptance of the paper before the rebuttal.  The authors provided suitable explanations in the rebuttal.  Overall, this paper advances the state of the art for MAP inference in credal networks.  This a solid work with good empirical and theoretical contributions.  Although some of the techniques are borrowed from the literature on Bayesian networks, the task of maximax and maximin MAP inference is new and it is the first work that proposes solutions for this task.

**Reviewer Concerns:**

The authors addressed the reviewers' concerns with suitable explanations.

Reviewer JLKE did not have any major concerns, but had several clarification questions that were addressed by the authors.  Hence, I expect the reviewer to maintain its score.

Reviewer Q5Sg raised the three concerns listed above.  The authors clarified that while some of the techniques were borrowed from the literature on Bayesian networks, it was the first time that they were utilized for credal networks.  Furthermore, the maximax and maximin tasks for MAP inference are new for credal networks.  The authors also clarified that the proposed techniques guarantee optimality with enough time.  The heuristics used are admissible, meaning that they still provide bounds that guarantee optimality.  While the heuristics are key to cut down the inference time, they do not change the worst case time complexity.  The paper does provide a time complexity analysis.  Finally, the claim of exact solution is not overstated.  The paper clearly explains that optimality is guaranteed only with enough time, which might be exponential.  While the paper mentions the use of heuristics that yield approximations, the heuristics are admissible and therefore still provide bounds that preserve optimality.  I expect the reviewer to maintain or raise its score.

Reviewer KRmC did not have major concerns, but had several clarification questions that were addressed by the authors.  Hence, I expect the reviewer to maintain its score.

Reviewer jvi6 only had minor concerns (typos and clarification) that were addressed by the authors.  Hence, I expect the reviewer to maintain its score.

**Reviewer Scores:**

I expect the reviewers to maintain their scores as explained above.

---

### Decision · Program_Chairs · 2026-01-26

Accept (Poster)